

# Reviews and syntheses: Systematic Earth observations for use in terrestrial carbon cycle data assimilation systems

Marko Scholze[1], Michael Buchwitz[2], Wouter Dorigo[3], Luis Guanter[4], and Shaun Quegan[5]

[1]Department of Physical Geography and Ecosystem Science, Lund University, Lund, Sweden
[2]Institute of Environmental Physics (IUP), University of Bremen, Bremen, Germany
[3]Department of Geodesy and Geoinformation, Vienna University of Technology (TU Wien), Vienna, Austria
[4]Remote Sensing Section, German Research Center for Geosciences (GFZ), 14473 Potsdam, Germany
[5]Centre for Terrestrial Carbon Dynamics, The University of Sheffield, Sheffield S3 7RH, U.K.

*Correspondence to:* M. Scholze (marko.scholze@nateko.lu.se)

**Abstract.**

The global carbon cycle is an important component of the Earth system and it interacts with the hydrological, energy and nutrient cycles as well as ecosystem dynamics. A better understanding of the global carbon cycle is required for improved projections of climate change including corresponding changes in water and food resources and for the verification of measures to reduce
anthropogenic greenhouse gas emissions. An improved understanding of the carbon cycle can be achieved by model-data fusion or data assimilation systems, which integrate observations relevant to the carbon cycle into coupled carbon, water, energy and nutrient models. Hence, the ingredients for such systems are a carbon cycle model, an algorithm for the assimilation, and systematic and well error-characterized observations relevant to the carbon cycle. Relevant observations for assimi-
lation include various in-situ measurements in the atmosphere (e.g. concentrations of $CO_2$ and other gases) and on land (e.g. fluxes of carbon water and energy, carbon stocks) as well as remote sensing observations (e.g. atmospheric composition, vegetation and surface properties).

We briefly review the different existing data assimilation techniques and contrast them to model
benchmarking and evaluation efforts (which also rely on observations). A common requirement for all assimilation techniques is a full description of the observational data properties. Uncertainty estimates of the observations are as important as the observations themselves because they similarly determine the outcome of such assimilation systems. Hence, this article reviews the requirements of data assimilation systems on observations and provides a non-exhaustive overview of current
observations and their uncertainties for use in terrestrial carbon cycle data assimilation. We report on progress since the review of model-data synthesis in terrestrial carbon observations by Raupach et al. (2005) emphasising the rapid advance in relevant space-based observations.





## 1 Introduction

The anthropogenic pertubation of the global carbon cycle has led to a global mean increase of 43%
in atmospheric $CO_2$ (from 280 ppm to 398 ppm) in 2014 compared to pre-industrial (before 1750)
levels (WMO, 2015), and is the main driver for climate change. The main causes for the increase in
$CO_2$ are burning of fossil fuels and land use change, which amount to emissions of $9.8 \pm 0.5$ GtC in
2014. However, only about 44% of these emissions stay in the atmosphere, the remainder is currently
taken up by the land biosphere ($\approx 30\%$) and the surface ocean ($\approx 26\%$) (Le Quéré et al., 2015). Pos-
itive climate-carbon cycle feedbacks, predominantly acting on land processes, may reduce this sink
capacity and thus accelerate global warming (Matthews et al., 2007). Also, the sink strength of the
terrestrial biosphere is more variable than that of the ocean (Ciais et al., 2013) and its quantification
by process-based terrestrial carbon cycle models exhibit large uncertainties (Le Quéré et al., 2015).

A common way to reduce uncertainties from process-based modelling is by confronting these
models with observational data. Raupach et al. (2005) pointed out that the systematic combination
of observational data with process modelling, which is commonly refered to as 'model-data fusion',
is an effective strategy for observing the Earth system. Model-data fusion, or more formally known
as data assimilation, is motivated by several benefits to make best use of observations and models
(Mathieu and O'Neill, 2008). These benefits include, among others, (1) forecasting and initialisation
(forward predictions in time based on past observations), (2) model and data quality control (regular
and systematic confrontation of model output with observations within their uncertainty statistics),
(3) combination of various data streams (combined constraints of independent observations can be
stronger than the sum of the individual constraints), (4) filling in regions with sparse observations
(consistent propagation of information from data-rich regions to data-poor regions), (5) estimating
unobservable quantities (through process-based relations in the model observations constrain mod-
elled quantities which are not directly measured) and (6) observing system design (what is the delta
of a new observation/instrument).

Systematic observations are a key ingredient for model-data fusion studies. Here, we focus on the
carbon cycle and the land-atmosphere system. The land-atmosphere components of the carbon cycle
are an important part of an integrated Earth observation system because of the close interactions on
land between the carbon cycle and the water and energy cycles, and hence its importance for climate
projections and climate change mitigation strategies through the monitoring and management of
terrestrial greenhouse gas sources and sinks.

Raupach et al. (2005) provide an analysis of the various elements of a Terrestrial Carbon Obser-
vation System (TCOS). The need, design and steps to be taken towards a TCOS were already out-
lined by others before (Cihlar et al., 2002; Global Carbon Project, 2003) but Raupach et al. (2005)
systematically reviewed two major components of a TCOS: the model-data fusion methods and
the observational data and data uncertainty characterisctics for some selected, main kinds of rele-
vant data. The requirements for a policy-relevant carbon observing system have been outlined by



Ciais et al. (2014). They review the current systematic carbon-cycle observations and illustrate the implementation of such a policy-relevant carbon observing system.

In this paper we provide an update of the observational data and data uncertainty characteristics as assessed by Raupach et al. (2005) with a focus on exisitng but also new and upcoming, relevant space-based observations (in the following referred to as Earth Observation (EO) data). In contrast

to Ciais et al. (2014), who focus on carbon-cycle observations, we focus here on any kind of relevant observational data to be (potentially) assimilated in a terrestrial carbon cycle data assimilation system (CCDAS). In a CCDAS the observations are used to constrain the underlying model (i.e. to move model output quantities closer to the observations and reduce their posterior uncertainties) usually by parameter optimisation. In that sense we are somewhat broader in terms of observed vari-

ables because also 'non-carbon' observations (such as soil moisture or land surface temperature) are able to constrain the carbon cycle indirectly through process information embedded in the underlying models. At the same time, the focus of our review is narrower than that of Ciais (2014), who also addressed ocean and anthropogenic components. Our focus lies on the terrestrial carbon cycle, because of the higher spatial and temporal variability in the net exchange fluxes and their associated

higher uncertainties than form the ocean and anthropogenic components.

The paper is organized as follows: in the next section we contrast data assimilation to recently established benchmarking activities and give a brief overview of commonly used data assimilation approaches and their applications in terrestrial carbon cycling. We continue with a short overview on data characteristics inlcuding an update on progress for some of the observations discussed in

Raupach et al. (2005). Since there has been much developments in the provision of remotely sensed observations we focus here on the characteristics of the most relevant EO data streams.

## 2 Model-data fusion

### 2.1 Data assimilation versus benchmarking

In the recent past the international land surface and terrestrial ecosystem modelling communi-

ties have recognized the importance of model benchmarking and evaluation (e.g. Luo et al., 2012; Foley et al., 2013). One of the reasons for this development is the huge range of model results from different models in key diagnostics of the land-atmosphere interface such as gross primary productivity (GPP) and latent heat flux (Prentice et al., 2015).

In general 'benchmarking' is understood as the quantification of performance against a reference

using some pre-defined metrics. The reference can either be output from some previous model simulations, other (ensembles of) models or reference datasets based on observations if the model simulates the analogue quantity. Luo et al. (2012) suggest a theoretical framework for benchmarking land models based on based on standardized references and metrics to measure model performance skills. A large variety of such metrics and their characteristics is introduced by Foley et al. (2013). Some



95 examples of benchmarking terrestrial carbon cycle models (either standalone or coupled to climate
models) are given by e.g. Randerson et al. (2009), Cadule et al. (2010) and Kelley et al. (2013).

The commonality between benchmarking/evaluation and data assimilation lies in the quantitative
assessment of model output. In benchmarking the quantitative assessment is performed by calculat-
ing some metrics against either observations or other references, while in data assimilation this is

100 achieved by defining a cost function, which quantifies the mismatch of some simulated model quan-
tity against observations weighted by their uncertainties (including a model uncertainty). But data
assimilation goes beyond benchmarking as it minimises the quantified mismatch to improve model
performance directly by adjusting either initial and boundary conditions, state variables and/or model
process parameters.

105 As pointed out by Prentice et al. (2015) there is a need for both model benchmarking and data
assimilation: Benchmarking as a routine application to improve confidence and evaluate the perfor-
mance (over time) in terrestrial carbon cycle modelling. However, if a benchmark test for a given
model fails this could simply imply that the model parameter values have not been specified correctly
and optmised against observations. In contrast, data assimilation, in particular when used for param-

110 eter optimisation, potentially identifies structural model and/or data deficiencies if the model-data
mismatch (or the benchmar test) is still inadequate after optimisation (see also Figure (1)).

### 2.2 Data assimilation methods

The general problem of model-data fusion, or, more strictly speaking, data assimilation can be for-
mulated like this: Given a model $M$, a set of observations $\mathbf{y}$ of some observables $\mathbf{o} = H(\mathbf{z})$, with

115 $\mathbf{z}$ being the state variables of the model and $H$ the observation operator, and prior information on
some target variables $\mathbf{x}$, produce an updated description of $\mathbf{x}$. $\mathbf{x}$ may include elements of $\mathbf{z}$ and $\mathbf{p}$
(parameter, quantities not changed by the model, i.e. process parameters, boundary and initial condi-
tions). Here, we follow the notation as introduced by Rayner et al. (2016). The observation operator
maps the model state onto observables. In the case of a CCDAS assimilating atmospheric $CO_2$ the

120 observation operator is the atmospheric transport model mapping the net $CO_2$ surface exchange
fluxes as calculated by the terrestrial carbon cycle onto simulated atmospheric $CO_2$ concentrations.
In transport inversions the dynamical model, the atmospheric transport model, is also the observation
operator.

A data-assimilation system consists of three main ingredients: a set of observations, a dynamical

125 model including the observation operator and an assimilation method. In the Bayesian formulation
of the assimilation problem uncertainties (i.e. the description of quantities by probability density
functions, PDFs) are central to the concept of data assimilation. Both observations as well as models
have errors arising for various reasons. We will detail the observational errors in the next section.
Dynamical models as well as observation operators have errors arising from the parameterizations,





and the discretization of analytical dynamics into a numerical model; for a more complete description
of uncertainty in Earth System models or components of such we refer to Scholze et al. (2012).

We distinguish two basic approaches in data assimilation: sequential assimilation, which assim-
ilates observations at discrete time-steps and thus evolves over time according to the dynamical
model; and variational assimilation, which assimilates all observations at once at their respective
measyurement time over a given period, the so-called assimilation window. They differ in their nu-
merical efficiency and optimality for their specific use. A general data-assimilation scheme is shown
in Figure (1). In the sequential approach the inner loop is evaluated sequentially over time following
the dynamics of the model. In the case of variational assimilation the inner loop is evaluated itera-
tively (assuming a non-linear model) until a cost function minimum is found. The cost function is
formulated as

$$J = \frac{1}{2}\left[(\mathbf{x} - \mathbf{x}^{\mathrm{b}})^T \mathbf{B}^{-1}(\mathbf{x} - \mathbf{x}^{\mathrm{b}}) + (\mathrm{H}(\mathbf{x}) - \mathbf{y})^T \mathbf{R}^{-1}(\mathrm{H}(\mathbf{x}) - \mathbf{y})\right], \tag{1}$$

where $\mathbf{x}^{\mathrm{b}}$ is the prior information, $\mathbf{B}$ the prior uncertainty covariance, and $\mathbf{R}$ the observational
uncertainty covariance. From Equation 1 follows that data and prior knowledge cannot be treated
separately from their respective uncertainties (Raupach et al., 2005). In other words, observations
(or prior knowlegde) for data assimilation are only complete if we know the full probability density
function (PDF), which, in the case of a Gaussian, can be characterised by its mean and variance. In
practical terms, the observational uncertainty covariances weight the model-data mismatch, while
the prior uncertainty covariances weight the deviation of the target variables from their prior values.
We note here that in the Gaussian case the model and observation operator errors can be added
quadratically to the observation errors.

An important diagnostic in data assimilation is the posterior uncertainty, which usually, because
of its high dimension, is hard to compute. If the assimilation problem is Gaussian the computation of
the posterior uncertainty covariance matrix simplifies and it can be approximated by the inverse of
the Hessian ($2_n d$ derivative) of the cost function. Typically, gradient-based optimisation approaches
approximate the Hessian, alternatively ensembles can be used to derive realisations of the poste-
rior PDF. The uncertainty reduction relative to the prior (i.e. $1 - \mathbf{U}_{\mathbf{x}\mathbf{po}}/\mathbf{B}$ with $\mathbf{U}_{\mathbf{x}\mathbf{po}}$ the posterior
uncertainty) then is a measure of the observational constraint on the target variables.

Rayner et al. (2016) introduce the theory fundamental to data assimilation and illustrate how the
different implementations of data assimilation relate to this theory in a more narrative style A more
complete and mathematically precise introduction to the concepts of data assimilation is given in the
textbooks by e.g. Daley (1991); Tarantola (2005).

### 2.3 Examples of terrestrial carbon cycle data assimilation

A variety of the methods as described by Rayner et al. (2016) have been applied by the carbon cy-
cle community. One example making use of formal assimilation methdologies for inferring surface-



atmosphere $CO_2$ exchange fluxes is based on atmospheric transport inversions. As mentioned before, in atmospheric inversions the observation model is an atmospheric tracer transport model. In atmospheric inversions both sequential and variational methods have been used together with observations of atmospheric trace gas concentrations such as from the flask sampling network, continuous in-situ and aircraft measurements and more recently also remotely sensed total column measurements. The

techniques for atmospheric transport inversions have been detailed in Enting (2002) and a recent comparison of results from different transport inversion is given by Peylin et al. (2013).

A more recent development is the assimilation of observations into terrestrial biosphere models. Here, various methods and observations have been used to optimise model process parameters at different scales. A comparison of a whole suite of these assimilation methods applied to a test case

using a simplified model at local-scale is given by Trudinger et al. (2007) and Fox et al. (2009).

Kaminski et al. (2002) were among the first who applied a formal algorithm together with observations of atmospheric $CO_2$ concentrations to constrain the Simple Diagnostic Biosphere Model at global scale. This work was continued by the development of the first Carbon Cycle Data Assimilation System with a process-based model at its core (Rayner et al., 2005). The advantage of using

a process-based model at the core of a CCDAS is that once the process parameters have been optimised the the constrained model can also be used for predictions as demonstrated by Scholze et al. (2007). Also, such systems are capable of ingesting multiple independent data streams besides atmospheric $CO_2$ concentrations. Kaminski et al. (2013) provide an overview on the developments of the CCDAS-BETHY since its first application while Scholze et al. (2016) demonstrate the latest ap-

plication of CCDAS-BETHY assimilating atmospheric $CO_2$ and remotely sensed surface soil moisture simultaneously. Since then several global terrestrial ecosystem models have been included in Carbon Cycle Data Assimilation Systems employing a variational approach (e.g. Schürmann et al., 2016; Peylin et al., 2016).

Concurrently, there have been several studies at the local/regional scale assimilating various types

of observations. For instance, Barrett (2002) used a genetic algorithm to infer soil carbon turnover times in a terrestrial carbon cycle model over Australia from plant production, biomass, litter and soil carbon observations. Local eddy covariance flux tower measurements of net exchange of $CO_2$ and latent and sensible heat fluxes have been assimilated to optimize parameter related to photosynthesis, respiration and energy fluxes of terrestrial ecosystem models, using Monte Carlo type methods

(e.g. Braswell et al., 2005; Knorr and Kattge, 2005; Moore et al., 2008; Ricciuto et al., 2008), sequential methods (Williams et al., 2005), as well as variational approaches (e.g. Wang et al., 2001; Kuppel et al., 2012; Raoult et al., 2016)

Recent advances are focusing on multiple independent data stream assimilation to provide a more rigorous constraint on the multiple components of terrestrial ecosystem models and avoid equifinal-

ity, i.e. different parameter solutions provide the same cost function value. Examples for such studies on local/regional scale are the assimilation of eddy covariance $CO_2$ fluxes together with observations



of vegetation structural information or carbon stocks (e.g. Richardson et al., 2010; Keenan et al., 2012) or together with remotely sensed vegetation activity such as the Fraction of Absorbed Photosynthetic Active Radiation (FAPAR) (e.g. Kato et al., 2013; Bacour et al., 2015). The assimilation

of multiple data streams can be performed either in a step-wise (e.g. Peylin et al., 2016) or simultaneous approach (e.g. Kaminski et al., 2012); in the case of non-linear models only the simultaneous assimilation makes optimal use of the observations (MacBean et al., 2016).

## 3   Data characteristics and provision

Observations are our measurable representation of the 'Truth'. They come with different charac-
teristics in terms of spatial and temporal resolution, coverage of the observed system, and errors. In analogy, models are also some representation of the 'Truth', however, via knowledge embodied in some form of functional relationships (with their own errors as mentioned before). The paper by Raupach et al. (2005) has been instrumental in highlighting the challenges in combining models and observational data for building a TCOS focussing on the observational requirements. Ciais et al.
(2014) argue for a globally integrated carbon observation system to improve our understanding of the carbon cycle for predicting future changes and to be able to independently verify the impact of emission reduction measures. Such a system relies on atmospheric carbon observations as a backbone but also concerns observations of the terrestrial and ocean carbon cycle. They focus on a strategy towards a global carbon-cycle monitoring system for achieving the above mentioned objectives.

Figure (2) depicts exemplarily the main observations of a TCOS and their space-time characteristics. In the following we briefly summarise the aspects of uncertainty in the observations and highlight progress on the specification of uncertainty for some of the observations in Fig. (2) as well as on their monitoring since Raupach et al. (2005).

### 3.1   Observational uncertainty

As mentioned before an important ingredient to any model-data fusion system are not only the observations themselves but also the uncertainties associated to them. We distinguish three main types of observation errors:

- Random: Random errors are always present in measurements and are caused by unpredictable changes in the measurement system (e.g. electronic noise in electrical instrument). They show
up as different readings of the same repeated measurement, and thus can be reduced by taking the average of multiple measurements. Random erros are usually assumed to be Normal (Gaussian) distributed, however, in some cases the random error distribution is log-normal (e.g. precipitation) or skewed by outliers due to unpredictable corruptions of the measurement system. Random erros are therefore relatad to the precision of a measurement system.



• Systematic (bias): Systematic errors in observations are usually due to some recurring prob-
        lems in the overall measurement system. They are caused by instrument miscalibrations or
        interferences with the measurement system. They can vary in space and time but they affect
        the measurement system in a predictable way. Biases can be both additive (absolute mean
        bias) and multiplicative (biases in the dynamic range affecting the amplitude of a signal). If
the source for systematic errors is known they can usually be fixed and shoud be removed.
        Systematic erros are therefore relatad to the accuracy of a measurement system.

      • Representativeness. The representation error occurs when information is represented at a scale
        different from the source of the information. For instance a quantity simulated by a model is
        'representative' for a given spatial and temporal resolution of the model grid. An individ-
ual measurement, however, represents information influenced by the local environment not
        resolved by the model grid (e.g. representation of atmospheric flask data in an atmospheric
        transport model gridcell). In the case of satellite-based observations the representation error
        also includes errors in inferring a biophysical quantity from the photons measured at the sen-
        sor. We come back to this issue later.

For both random and systematic errors not only the magnitude of the error for a single observation
is important, i.e. the diagonal elements in the observational uncertainty covariance matrix $\mathbf{B}$, but also
the correlations among errors for different observations. Hence there is a need to specify the off-
diagonal elements in the error covariance matrix $\mathbf{B}$. These off-diagonal elements are usually hard to
specify, however, they are important to quantify in a data assimilation system because they affect the
prediction of the optimal solution in the same way as the diagonall elements.

As mentioned before, Raupach et al. (2005) have already reflected on the main properties of the
data and their error covariances for observations of remotely sensed land surface properties (mainly
NDVI), atmospheric $CO_2$ concentrations, land-atmosphere net $CO_2$ exchange fluxes, and terres-
trial carbon stores. In the past years, there has been substantial progress in the specification of un-
certainties in eddy-covariance measurements of the land-atmosphere net $CO_2$ exchange flux (Net
Ecosystem Productivity, NEP) and its component fluxes (GPP and ecosystem respiration, $R_{eco}$). For
instance, Lasslop et al. (2008) analysed the error distribution and found that the eddy flux data can
almost entirely be represented by a superposition of Gaussian distributions with inhomogeneous
variance. In a more recent study Raj et al. (2016) investigated the uncertainy of GPP derived from
partitioning the eddy covariance NEP measurements. They used a light-use efficiency model em-
bedded in a Bayesian framework to estimate the uncertainty in the separated GPP from the posterior
distribution at half-hourly time steps. The availability of the eddy covariance data has also been
heavily improved; the latest release of the FLUXNET2015 dataset now contains data from about
165 sites worldwide spanning a period from 1991 (for some sites) up to 2014 (FLUXNET2015).



### 3.2 Towards operational carbon observation systems

In the European framework there have recently been major developments towards systematic in-situ observations for use in terrestrial carbon cycle data assimilation systems. The Integrated Carbon Observing System (ICOS) is a novel pan-European infrastructure for carbon observations, which will provide high-quality in-situ observations (both fluxes as well as atmospheric concentrations) over Europe and over ocean regions adjacent to Europe with a long-term perspective. ICOS consists of central facilities for co-ordination, calibration and data in conjunction with networks of atmospheric, oceanic and ecosystem observations as well as a data distribution centre, the Carbon Portal, providing discovery of and access to ICOS data products such as derived flux information. The ICOS network runs in an operational mode, and greenhouse gas concentrations and fluxes will be determined on a routine basis. The measurements are designed to allow up to daily determination of (mainly natural) sources and sinks at scales down to approximately 50 x 50 $km^2$ for the European continent.

An example for an operationalised, space-based Earth observing programme is the fleet of so-called Sentinel satellites of the European Copernicus programme. Copernicus aims at providing Europe with a continuous and independent access to Earth observation data and associated services (transforming the satellite and additional in-situ data into value-added information by processing and analysing the data) in support of Earth System Science (Berger et al., 2012). So far, six different Sentinel missions are planned out of which three are in operation and the remainder is scheduled to be launched during the next years. Each type of the currently foreseen Sentinels has a specific objective and will deliver a range of EO products. Some of these products will be suitable for constraining the terrestrial carbon cycle, such as soil moisture (Sentinel 1), FAPAR, leaf chlorophyl and water content and land cover (Sentinel 2 and 3), land surface temperature (Sentinel 3), atmospheric methane and flourescence (Sentinel 5 and precursor). So far, a dedicated mission for monitoring the carbon cycle, i.e. an instrument measuring the atmospheric $CO_2$ composition, is not yet included in the Copernicus monitoring programme (see Ciais et al., 2015), however, the series of Sentinel satellites is likely to be extended in the future.

### 3.3 Examples of systematic observations from satellite EO data

There has been a vast extension of EO capabilites during the past 10 years or so both in terms of product quality (including, for instance, improved accuracy) but also quantity (new products).

In any data assimilation system using satellite EO data one needs to decide in the design phase of the system whether to assimilate observations at the sensor level (i.e. the spectral radiances for optical sensors or brightness temperatures for microwave sensor, referred to as level 1 data) or to assimilate the bio-geophysical variable derived from the radiances through a retrieval algorithm (level 2 data product). When assimilating level 1 data the retrieval algorithm is part of the observation operator linking the model state to the observations in the data assimilation system. A more detailed



305   description of the two alternatives in assimilating EO satellite observations into models of the Earth
system is given by Kaminski and Mathieu (2016). In carbon cycle data assimilation systems level
2 data products (or even level 3 data, which are provided on a regular space-time grid) are most
commonly used.

In the next subsections we present some selected, and for terrestrial carbon cycle data assimi-
lation most relevant remotely sensed Earth Observation products in more detail. The EO products
described below (atmospheric $CO_2$, vegetation activity, soil moisture, terrestriaial biomass) either
have already been used, are in the process of being used, or would potentially be a useful data
constraint in a CCDAS. For vegetation activity we distinguish two major types of products: more
'traditional' reflectance- or radiative-based products such as fraction of absorbed photosynthetically
active radiation (FAPAR) and recently developed products based on biogeochamical processes such
as sun-induced flourescence (SIF). For instance, FAPAR has already been demonstrated to provide
a strong constraint on terrestrial carbon as well as water fluxes through its impact on the phenol-
ogy components of the carbon cycle model (e.g. Knorr et al., 2010; Kaminski et al., 2012). SIF is a
promising observation to constrain the gross uptake of $CO_2$ by plant photosynthesis. First assimi-
lation results using SIF observations in a CCDAS show that the uncertainty in global annual GPP
is largely reduced by constraining parameters that describe leaf phenology (Norton et al., 2016).
Also assimilation of XCO2 into a diagnostic terrestial carbon cycle model has been shown to derive
net $CO_2$ fluxes consistent with independent in-situ measurements of atmospheric $CO_2$ as well as
to reduce posterior uncertainties in the inferred net and gross $CO_2$ fluxes (Kaminski et al., 2016b).
van der Molen et al. (2016) assessed the impact of assimilating various remotely sensed soil mois-
ture products into the SiBCASA ecosystem model on simulated carbon fluxes in Boreal Eurasia.
Although the impact of assimilating ASCAT surface soil moisture was significant, its skill in this
hydrologically complex environment strongly depends on surface water and vegetation dynamics. In
contrast, Scholze et al. (2016) showed that when assimilating SMOS soil moisture simultaneously
with in-situ atmospheric $CO_2$ concentrations the reduction of uncertainty in gross and net $CO_2$ fluxes
relative to the prior is considerably higher than for assimilating $CO_2$ only, which quantifies the added
value of SMOS observations as a constraint on the terrestrial carbon cycle. So far, remotely sensed
biomass data have not been used in carbon cycle data assimilation studies, however, Thum et al.
(2016) demonstrated the added value of in-situ observations of biomass increment in reducing un-
certainties in simulated above ground biomass mainly through the calibration of parameters in the
carbon allocation scheme of the terrestrial carbon cycle model.

This list of EO products described in this paper is admittedly subjective and there is of course a
whole range of additional remotely sensed products available, which are relevant for carbon cycle
studies as well, e.g. burned area (e.g. Giglio et al., 2013), land cover (e.g. Bontemps et al., 2012),
land surface temperature (e.g. Li et al., 2013), leaf area index (which is in effect closely related to
FAPAR) (e.g. Liu et al., 2014) or vegetation optical depth (e.g. Konings et al., 2016). However, these



products are rather used as input or boundary conditions for terrestrial carbon cycle models or, for instance in the case of vegetation optical depth, they have so far not been used in carbon cycle data assimilation studies.

### 3.3.1 Atmospheric $CO_2$ and $CH_4$


Satellite retrievals of atmospheric carbon dioxide ($CO_2$) and methane ($CH_4$) are available from several satellite instruments such as mid-tropospheric $CO_2$ and $CH_4$ columns from Infrared Atmospheric Sounding Interferometer (IASI) (e.g. Crevoisier et al., 2009a, b) on EUMETSAT's Metop satellite series, vertical profiles with highest sensitivity in the mid/upper troposphere from AIRS

on Aqua (e.g. Xiong et al., 2013), stratospheric profiles from MIPAS on ENVISAT limb observations (e.g. Laeng et al., 2015) and from the solar occultation observations of SCIAMACHY on ENVISAT (Noël et al., 2011, 2016) and ACE-FTS (e.g. Boone et al., 2005; Foucher et al., 2009). These observations have however only little or no sensitivity to $CO_2$ and $CH_4$ concentration changes close to the Earth's surface and therefore contain only limited information on regional or local

$CO_2$ and $CH_4$ sources and sinks. Satellites with high near-surface sensitivity are nadir (downlooking) satellites which measure radiance spectra of reflected solar radiation in the relevant spectral bands in the near-infrared/shortwave-infrared (NIR/SWIR) spectral region, which are located around 1.6 $\mu$m ($CO_2$ and $CH_4$) and around 2.0 $\mu$m ($CO_2$). Satellites instruments which perform (or have performed) these observations are SCIAMACHY onboard ENVISAT (2002–2012) (Burrows et al.,

1995; Bovensmann et al., 1999), TANSO-FTS onboard GOSAT (launched in 2009) (Kuze et al., 2009, 2014) and NASA's Orbiting Carbon Observatory 2 (OCO-2) mission (launched in 2014) (Crisp et al., 2004; Boesch et al., 2011).

The main $CO_2$ and $CH_4$ data products of these sensors are near-surface-sensitive column-averaged dry-air mole fractions of $CO_2$ and $CH_4$, denoted XCO2 and XCH4. The quantities XCO2 and XCH4

are both retrieved from SCIAMACHY/ENVISAT (ground pixel size: $30\times50$ km$^2$ (along track times across track); swath width 960 km with contiguous ground pixels) and TANSO-FTS/GOSAT (10 km pixel size; several (e.g. 3 or 5) non-contiguous pixels across track). OCO-2 delivers XCO2 (8 ground pixels across track, each $\approx$1.3 km) and in the near future other satellites will be launched such as Europe's Sentinel-5-Precursor satellite (S5P) (Veefkind et al., 2012), which will deliver (among

several other parameters) XCH4 (7 km pixel size at nadir, 2600 km swath width with contiguous ground pixels; planned launch: mid 2017) (Butz et al., 2012) and China's TanSat (planned launch end of 2016), which will deliver XCO2 with similar characteristics as NASA's OCO-2. In the following we will focus the discussion on sensors who have already delivered multi-year XCO2 and XCH4 year data sets, i.e. on SCIAMACHY and TANSO.

The XCO2 and XCH4 data products retrieved from SCIAMACHY and TANSO are generated from the radiance observations using different approaches. Most approaches are based on 'Optimal Estimation' (OE) (e.g. Rogers, 2000; Reuter et al., 2010), also called Bayesian inference.



OE permits to constrain the retrieval using a priori information on, e.g. atmospheric vertical pro-
files of trace gases and aerosols. In general, the radiances are simulated using a radiative transfer
model (RTM) and RTM and other parameters (state vector elements) are adjusted until an 'opti-
mal' match is achieved between observed and simulated radiances. One algorithm (WFM-DOAS
(WFMD) (Buchwitz et al., 2000; Schneising et al., 2008, 2009)) is based on least-squares and does
not use a priori information to constrain the fit parameters. As a consequence, the resulting XCO2
and XCH4 products are typically somewhat 'noisier' compared to the OE products.

The XCO2 and XCH4 data products from SCIAMACHY are generated within the GHG-CCI
project (Buchwitz et al., 2015) of ESA's Climate Change Initiative (CCI, Hollmann et al. (2013))
and these products are available from the GHG-CCI website (http://www.esa-ghg-cci.org/). XCO2
and/or XCH4 products from GOSAT are generated at several institutions in Japan, Europe and the
USA and are available from several sources as shown in Table (1). The quality of these GHG-CCI
products and the XCO2 and XCH4 products generated elsewhere has been significantly improved
during recent years (e.g. Schneising et al., 2012; Yoshida et al., 2013; Dils et al., 2014; Buchwitz et al.,
2015) and has now reached quite high maturity when compared to user requirements as formulated
in, e.g. GCOS (2011). This can be concluded, for example, from the quality of the latest version
of the GHG-CCI SCIAMACHY and TANSO XCO2 and XCH4 data set ('Climate Research Data
Package No. 3', CRDP3) (Buchwitz et al., 2016). Based on comparisons with ground-based obser-
vations of the Total Carbon Column Observing Network (TCCON, Wunch et al. (2010, 2011)) it
has been found that the GCOS requirements for systematic error (< 1 ppm for XCO2, < 10 ppb
for XCH4) and long-term stability (< 0.2 ppm/year for XCO2, < 2 ppb/year for XCH4) are met for
nearly all products. As also shown in Buchwitz et al. (2016), the single observation (ground pixel)
retrieval precision) random error primarily due to instrument noise) is about 2 ppm for XCO2 from
SCIAMACHY and TANSO and ≈15 ppb for TANSO XCH4. For SCIAMACHY XCH4 the pre-
cision depends on time period and retrieval algorithm and is in the range 35 - 80 ppb. For some
products it has also been investigated to what extent the uncertainty can be reduced upon averaging
(Kulawik et al., 2016) and recommendations are given how to take into account error correlations
(Reuter et al., 2016), i.e. which values to use for the non-diagonal elements of the error covariance
matrix, as an important contrbution to the full characterisation of the data needs for data assimilaton
studies.

Figure (3) presents an overview about GHG-CCI CRDP3 XCO2 (left) and XCH4 (right) data set
in terms of time series and maps. These figures have been generated by gridding the underlying in-
dividual ground pixel (Level 2) products to generate a 5°×5° monthly Level 3 'Obs4MIPs' product
Buchwitz and Reuter (2016). Each 5°×5° monthly grid cell also contains an estimate of the over-
all uncertainty (also shown in Fig. (3)) which has been computed taking into account random and
systematic error components. As can be seen from Fig. (3), the uncertainty of the satellite XCO2
retrievals for monthly 5°×5° averages is estimated to be typically around 0.5 - 1 ppm (values larger




than 1 ppm are typically associated with regions where only few observations per grid cell exist,
e.g. due to clouds or higher latitudes corresponding to low sun elevation). For XCH4 the uncertainty
is on the order of a few ppb (typically 4 - 8 ppb). In Buchwitz and Reuter (2016), also initial TCCON
validation results of the Obs4MIPs products are presented. It is shown that the XCO2 product agrees
with monthly averaged TCCON XCO2 within $0.29 \pm 1.2$ ppm ($1\sigma$) and the XCH4 product within

$2.0 \pm 10.7$ ppb. This is hardly worse that the results which have been obtained by careful valida-
tion of the individual ground pixel retrievals taking into account the best possible spatio-temporal
co-location and considering the averaging kernels, etc. (e.g. Buchwitz et al., 2016). Note that the
computed differences of Obs4MIPs monthly $5° \times 5°$ satellite products with monthly averaged TC-
CON include the errors of the satellite data, errors of the TCCON products, errors due to neglecting

altitude sensitivity differences (averaging kernels), and representativity error. This indicates that the
representativity error is quite small (at least for monthly $5° \times 5°$ spatio-temporal sampling and res-
olution), probably on the order of 0.1 - 0.2 ppm for XCO2 and a few ppb for XCH4 (it is planned
to quantify this error in the future but currently only these rough estimates are available). Note that
detailed information on all GHG-CCI products is available on the GHG-CCI website in terms of

technical documents, links to peer-reviewed publications and figures including detailed maps for
each month and each individual data product.

    The SCIAMACHY and TANSO XCO2 and XCH4 retrievals have been used in a number of scien-
tific studies to address important questions related to the sources and sinks of atmospheric $CO_2$ and
$CH_4$ by atmospheric inversion studies (e.g. Bergamaschi et al., 2013; Houweling et al., 2015) and

more recently also in a data assimilation context for optimising model parameters (Chevallier et al.,
2016). Obviously, the longer the time series and the more accurate it is, the larger the information
content of a given data set. Therefore, further improvements are desired (Chevallier et al., 2016) and
possible (at least in terms of time series extension but likely also in further reduction of remaining
biases).

**3.3.2   Reflectamce-based vegetation dynamics/activity**

Since the early beginnings of remote sensing the state and evolution of the vegetation has been
monitored by satellites. An early attempt to analyse vegetation dynamics from space is to calculate
the Normalized Difference Vegetation Index (NDVI, defined as the ratio between the difference of
near-infrared, NIR, and visible red, Red, spectral bands and the sum of NIR and Red: NDVI = (NIR

- Red)/(NIR + Red), Deering et al. (1975)). The advantage of an index such as NDVI lies in its
simplicity and applicability to sensors with few spectral bands such as the Advanced Very High Res-
olution Radiometer (AVHRR). Therefore this index has been applied for numerous purposes over
the last 30 years or so. But NDVI is not a geophysical variable and it is sensitive to various perturb-
ing factors such as atmospheric constituents (aerosols, water vapor), directional effects (geometry

of illumination and observation), changes in soil background color changes (depending on soil wa-





ter content)(e.g. Pinty et al., 1993; Goel and Qin, 1994; Leprieur et al., 1994; Dorigo et al., 2007). There have been many attempts in modifying NDVI and developing additional vegetation indices to overcome its limitations, for example: Soil-Adjusted Vegetation Index (Huete, 1988), Atmospherically Resistant Vegetation Index (Kaufman and Tanre, 1992) or Global Environmental Monitoring

Index (Pinty and Verstraete, 1992). These indices generally exhibit some improvement in one respect but at the expense of some degradation in another respect.

A rational approach to address all these issues at once is to design a physically based quantity which closely follows the state of the vegetation. The Fraction of Absorbed Photosynthetically Active Radiation (FAPAR) provides some kind of information on the photosynthetic activity of the

land vegetation. FAPAR is recognised as an Essential Climate Variable (ECV) (GCOS, 2011) and is based on the land surface radiation budget. It is defined as the fraction of the photosynthetically active radiation (i.e. incoming solar radiation in the spectral region 0.4–0.7$\mu$m) that is absorbed by the vegetation canopy (see also Pickett-Heaps et al. (2014) for a mathematical definition). Several FAPAR products are derived from a variety of optical sensors (e.g. ATSR, MERIS, MISR, MODIS, SEVIRI,

SeaWiFS, VEGETATION) at different spatial and temporal resolutions. Although there has been substantial efforts to harmonize products across sensors (Ceccherini et al., 2013) and establish standards and validation practices (e.g. Widlowski, 2010) there are still considerable differences among the products. These differences can mainly be associated to differences in the retrieval methodology as well as to the quality of input variables. A recent overview of various FAPAR products and their

specifications, but without an assessemt of product uncertainties, is given by Gobron and Verstraete (2009). Table (2) summarises the characteristics of the most common FAPAR products.

Several studies have compared the performance of different satellite-derived FAPAR products: McCallum et al. (2010) looked at four FAPAR datasets over Northern Eurasia for the year 2000, Pickett-Heaps et al. (2014) evaluated six products across Australia, D'Odorico et al. (2014) com-

pared three products over Europe and Tao et al. (2015) assessed five products over different land cover types. Pickett-Heaps et al. (2014) concluded that although all six evaluated products display robust spatial and temporal patterns there is considerable disagreement amongst the products and non of the products outperforms the others. One of the reasons for these differences are different assumptions on the underlying biome types. They also reviewed the consistency of the FAPAR

products against in-situ field measurements, the mean difference between the EO products and the in-situ field measurements is around 0.1 (as FAPAR is a normalised fraction values range from 0 to 1). This estimate is confirmed by the study of Tao et al. (2015) who suggest an average uncertainty of 0.14 from validation against total FAPAR and 0.09 from validation against green FAPAR in-situ measurements. In their comparison of Joint Research Centre–Two-stream Inversion Package (JRC-

TIP) MODIS, JRC MGVI and Boston University MODIS products (see Tab. 2) D'Odorico et al. (2014) placed special emphasis on the assessment of the product uncertainties by not only comparing the uncertainties (or quality indicators) as proposed by the product teams but also by calculating



an independent theoretical uncertainty based on the triple collocation (TC) method (see Sec. 3.3.4).
While the uncertainties specified by the product teams differed by up to 0.1 among the products, the
TC method suggested more consistent uncertainties among the three products of around 10-20% of
the signal.

The JRC-TIP (Pinty et al., 2007) is an inverse modelling system that was deliberately designed
to retrieve a set of land surface variables, including FAPAR, in a form that is compliant with the
requirements for assimilation into terrestrial biosphere models. TIP is based on a one-dimensional
two-stream representation of the radiative transfer in the canopy-soil system (Pinty et al., 2006) and
applies the same inversion approach as CCDAS, which is briefly sketched in 2.2 and detailed in
Rayner et al. (2016); Kaminski and Mathieu (2016). In a first step it retrieves a set of model param-
eters describing the state of the vegetation canopy system including their full uncertainty covariance
by combining prior information with observed radiant fluxes. Further, the model is used to propagate
this PDF forward onto the simulated fluxes such as FAPAR. TIP uses observed broadband albedo
in the NIR and visible spectral domains as input from which it retrieves the effective (i.e. model-
dependent) quantities such as FAPAR, leaf area index (LAI) besides other radiatitve quantities.
Long-term global records of JRC-TIP products (see 2) have been retrieved from broadband albe-
dos provided by MODIS collection 5 (Pinty et al., 2011b, c) and Globalbedo (Disney et al., 2016).
Products are provided for each of the respective 16-day (MODIS) and 8-day (Globalbedo) synthesis
periods. Both JRC-TIP records are provided in the native 1 km resolution of the albedo input prod-
ucts. In order to maintain the above-mentioned compliance with terrestrial models, coarser resolu-
tion products are to be derived by applying JRC-TIP to aggregated albedo inputs (as in Disney et al.,
2016). JRC-TIP products are validated at site (Pinty et al., 2007, 2008, 2011a) and regional scales
(Disney et al., 2016); more details on JRC-TIP are given in Kaminski et al. (2016a).

### 3.3.3   Biogeochemical-based vegetation activity

Sun-induced fluorescence (SIF) is an electromagnetic signal emitted as a two-peak spectrum be-
tween 650 and 850 nm by the chlorophyll$-a$ of green plants under solar radiation. SIF can be di-
rectly related to photosynthetic electron transport rates and yields a mechanistic link to photosyn-
thesis and the subsequent gross carbon uptake by terrestrial vegetation (GPP) (Porcar-Castell et al.,
2014). Recent developments in satellite-based spectroscopy have enabled the first retrievals of SIF
from space (Frankenberg et al., 2011c; Joiner et al., 2011), which holds the promise of enabling new
approaches to globally monitoring terrestrial photosynthesis. For example, a high linear correlation
between data-driven GPP estimates and SIF retrievals at global and annual scales was reported by
Frankenberg et al. (2011c); Guanter et al. (2012). The skills of SIF as a proxy for photosynthetic ac-
tivity and GPP were also reported by studies over different ecosystems, like the Amazon rainforest
(Lee et al., 2013; Parazoo et al., 2013), large crop belts (Guanter et al., 2014), and the boreal forests
in Eurasia and North America (Walther et al., 2015).



The global retrieval of SIF from space lies on the principle of *in-filling* of solar Fraunhofer lines by SIF (Frankenberg et al., 2011b). Fraunhofer lines are absorption features in the solar spectrum, caused by elements in the solar atmosphere and sufficiently resolved by atmospheric spectrometers. Because of the additive nature of SIF, the fractional depth of the Fraunhofer lines detected by the satellite instrument decreases with the amount of SIF being emitted at the same wavelength. The retrieval of SIF from space is then based on the evaluation of the depth of the Fraunhofer lines present in red and NIR top-of-atmosphere spectra. The retrieval forward model is thus simple and can be linearised (e.g. Guanter et al., 2012; Köhler et al., 2015b), so the inversion can be easily solved by least squares optimisation.

Fraunhofer line-based SIF retrievals tend to be accurate but not precise: uncertainties are dominated by a random component associated to instrumental noise, which is linearly mapped into SIF retrievals. The amplitude of instrumental noise, and hence 1-$\sigma$ single-retrieval errors, scale with at-sensor radiance for the most common case of grating-based spectrometers dominated by multiplicative noise. This implies that retrieval errors are mostly driven by surface brightness and sun zenith angles (Guanter et al., 2015). Because of this high contribution of random errors to the total retrieval uncertainty, single SIF retrievals are commonly linearly-aggregated as spatio-temporal composites in which random errors are reduced. The amount of retrievals to be aggregated into a given gridbox results from a compromise between spatial resolution, temporal resolution and precision of the gridded product. The random uncertainty of the resulting spatio-temporal composites is then not only driven by surface albedo and illumination, but also by the number of soundings going into a given gridbox, which is in turn defined by cloudiness and latitude (in the case of overlapping orbits). Detailed analyses of random errors in SIF retrievals for different spaceborne instruments can be found in Frankenberg et al. (2011b) and Guanter et al. (2015).

Global SIF data sets have been or are being derived from GOSAT, MetOp's Global Ozone Monitoring Experiment-2 (GOME-2), ENVISAT's SCIAMACHY and the OCO-2 mission (Joiner et al., 2011; Frankenberg et al., 2011c; Guanter et al., 2012; Joiner et al., 2012, 2013; Köhler et al., 2015a, b; Wolanin et al., 2015; Joiner et al., 2016; Frankenberg et al., 2014). Sample SIF maps from GOSAT, GOME-2 and SCIAMACHY for July 2010 are displayed in Fig. 4. All four missions except for SCIAMACHY are still operating. The spectal, spatial and temporal sampling of single SIF soundings varies for each instrument, as it is summarised in Table 3. For example, GOME-2 and SCIAMACHY provide SIF retrievals in the red and NIR spectral regions with global coverage and a relatively high temporal resolution. However, this comes at the expense of a coarse spatial resolution, which is $40 \times 80\,\mathrm{km}^2$ for GOME-2 ($40 \times 40\,\mathrm{km}^2$ for GOME-2 on MetOp-A since July 2013) and $30 \times 240\,\mathrm{km}^2$ for SCIAMACHY. On the other hand, GOSAT and OCO-2 do not provide spatially-continuous measurements (i.e. no global coverage), but single soundings in the NIR have a much higher spatial resolution than those of GOME-2 and SCIAMACHY. In particular, OCO-2 soundings correspond to ground areas of about $4\,\mathrm{km}^2$, which is substantially finer than that of the other data





sets. The number of soundings per day by OCO-2 is also much larger (about 100x) than that by the other instruments (Frankenberg et al., 2014), which makes OCO-2 SIF to be the most suited data set for studies over areas not requiring a continuous spatial sampling but benefiting from a high spatial resolution. This is the case, for example, of tropical and boreal forests: spatial continuity is less criti-

cal for those ecosystems because they are relatively homogeneous over large spatial scales, whereas the high spatial resolution is important to maximise the number of clear-sky soundings during the parts of the year with persistent cloudiness.

Concerning near-future perspectives for SIF monitoring, it can be expected that the limitations in spatial resolution and coverage of existing SIF products will be alleviated with the advent of the

TROPOspheric Monitoring Instrument (TROPOMI) scheduled for launch onboard the Sentinel-5 Precursor satellite mission by mid 2017 (Table 3). TROPOMI will enable SIF retrievals in the red and NIR regions similar to GOME-2 and SCIAMACHY, but with a 7 km pixel, daily global coverage and a number of clear-sky observations per day ≈200 larger than GOME-2 and ≈600 larger than SCIAMACHY. The SIF product from TROPOMI can therefore be anticipated to have a much higher

spatio-temporal resolution and signal-to-noise ratio than those from GOME-2 and SCIAMACHY (Guanter et al., 2015). Complementary, the FLuorescence EXplorer (FLEX) (Drusch and FLEX Team, 2015) has recently been selected for implementation by ESA, with launch currently expected for 2022. FLEX will provide global measurements of SIF in the red and NIR with at a relatively low temporal resolution, but with the finest spatial resolution of all existing and upcoming spaceborne

instruments.

### 3.3.4   Soil moisture

Soil moisture is measured in-situ through large-scale soil moisture monitoring networks (Dorigo et al., 2011; Ochsner et al., 2013) or at various FLUXNET sites (Baldocchi et al., 2001). Yet, these point observations have only limited coverage in space time, have spatially very divergent properties

(Dorigo et al., 2013), and often contain large representativeness errors at the scale of global ecosystem models (Gruber et al., 2013). Satellite remote sensing in the microwave domain has the potential to overcome many of these issues. Microwave remote sensing uses the contrasting dielectric properties of water, air, ice, and soil particles to infer the water content in the soil column (Owe et al., 2008). Both passive radiometer systems, measuring the emitted microwave radiance ('brightness

temperatures'), and active radar systems, measuring backscattered microwave radiance, can be used to retrieve soil moisture. Microwave sensors operate in different frequency (wavelength) domains, of which L-band (with a wavelength of ≈23 cm) and C-band (≈5 cm) are most commonly used for retrieving soil moisture (Kerr et al., 2012; Owe et al., 2008; Wagner et al., 1999). Smaller wavelengths are more sensitive to the vegetation canopy covering the soil and increasingly lose their sensitivity

to water. Still, frequencies up to 19 GHz (≈1.5 cm) have proven potential for providing robust soil moisture estimates at the global scale for moderately to sparsely vegetated areas (Owe et al., 2008).



Due to the relatively low energy levels and the technical challenges in microwave domain, spatial resolutions of the satellite observations are generally coarse ($\approx$25–50 km) but with high revisit frequencies (up to 1 day). Only Synthetic Aperture Radar is able to provide much higher spatial resolutions up till a few meters, yet at the cost of the revisit times.

Since the release of the first global soil moisture datasets from microwave sensors in the early 2000s the number of available soil moisture products and missions has rapidly expanded (De Jeu and Dorigo, 2016). Several (pre-)operational products are now available from a wide variety of data providers and space organizations (Table 4). While initially soil moisture products were based on sensors mainly designed for other purposes (such as ASCAT, AMSR2, and Sentinel-1), ESA and NASA launched their own dedicated soil moisture satellite missions SMOS and SMAP (Kerr et al., 2012; Entekhabi et al., 2010). Apart from the Sentinel-1 mission, which primarily targets the provision of high resolution observations over Europe, all currently active missions provide a nearly global coverage at a coarse resolution approximately every 1-2 days. Differences between the various products exist in their technical design, observation bands, and retrieval algorithms, which often result in complementary strengths over different land cover types (Alyaari et al., 2015; Dorigo et al., 2010; Liu et al., 2011). The missions also differ in their degree of operationalization: While SMOS and SMAP are primarily scientific concept demonstrators, AMSR2 continues the legacy of C-band radiometer observations started by JAXA and NASA in 2002 with the launch of AMSR-E, while AS-CAT is embedded in a fully operational program of weather observing satellites with a guaranteed continuation at least until 2023 and a follow-on mission already under development (Wagner et al., 2013). Apart from the target variable surface soil moisture, some products come with estimates of freeze/thaw state and vegetation optical depth, which are disentangled from the soil moisture impacts on the measured microwave signal during the retrieval process.

As none of the currently active missions covers a period long enough to study climate change impacts, ESA's Climate Change Initiative (CCI) endorsed the combination of available soil moisture products from active and passive microwave sensors into a consistent multi-decadal record. The ESA CCI soil moisture product currently combines soil moisture products from 11 different sensors into a homogenized daily product spanning the period 1978-2015 (Liu et al., 2012, 2011; Dorigo et al., 2016). Several studies have demonstrated the value of ESA CCI soil moisture for assessing long-term interactions between soil moisture and vegetation productivity (Barichivich et al., 2014; Chen et al., 2014; Dorigo et al., 2012; Muñoz et al., 2014).

Key to a proper assimilation of remotely sensed soil moisture into carbon models is a correct characterization of its errors. Apart from instrument errors which are common to all observations, the quality of microwave-based soil moisture retrievals is particularly impacted by vegetation cover, soil frost, snow cover, open water, topography, surface roughness, urban structures, and radio frequency interference (Dorigo et al., 2010; Kerr et al., 2012). Observations where a strong adverse impact of these factors is detected are usually masked during processing, which may lead to data gaps for



certain areas or periods of the year (Dorigo et al., 2015). If cases where their impact on the soil
moisture retrieval is only moderate, the errors that they introduce are either simulated during the
retrieval itself using error propagation methods, or assessed a posteriori against reference data using
various statistical methods (Draper et al., 2013).

While the ASCAT and AMSR2 products come with an estimate of the error variance for each ob-
servation by error propagation (Naeimi et al., 2009; Parinussa et al., 2011) this is still not common
practice for all soil moisture products. Yet, no error propagation model perfectly represents all error
sources and interactions (Draper et al., 2013). On the other hand, the use of in-situ soil moisture
measurements to estimate random errors is hampered by their heterogeneous nature and large spatial
representativeness errors (Gruber et al., 2013). As an alternative, in recent years triple collocation
analysis (TCA) has firmly established itself as a robust alternative to estimate random errors in soil
moisture datasets without the need of an absolute 'true' reference (Dorigo et al., 2010; Scipal et al.,
2008). TCA estimates the error variances of three spatially and temporally collocated soil moisture
datasets with independent error structures, e.g. a radiometer-based, a scatterometer-based, and a land
surface model soil moisture dataset. Recently, the TCA has been intensively elaborated, e.g. to solve
for collinearities between errors (Gruber et al., 2016b) and non-linear dependencies between datasets
(Zwieback et al., 2016). The most remarkable advancement has been to express TCA-based error es-
timates as a signal-to-noise ratio, which facilitates a direct intercomparison of the skill of datasets in-
dependent of their dynamic ranges (Gruber et al., 2016a), 5. Although the TCA provides an estimate
that is entirely independent of any retrieval model assumptions, it only provides a single average er-
ror estimate for the entire period under consideration. Thus, synergistic use of error propagation and
triple collocation estimates shall be exploited to better capture the temporal error dynamics needed
for an optimal assimilation into carbon models. Due to the recent progress in product quality, er-
ror characterization, and operationalization, satellite-based soil moisture products have reached the
level of maturity that allows for a systematic assimilation into land surface models to improve the
models' hydrology. For example, Martens et al. (2016) showed that the assimilation of SMOS and
ESA CCI soil moisture generally has a small positive impact on soil water storages and evaporative
fluxes as simulated by the GLEAM land evaporation model. Surface soil moisture from ASCAT
is assimilated operationally in near-real-time into ECMWF Land Data Assimilation System to ob-
tain root-zone soil moisture (Albergel et al., 2012). Global root-zone soil moisture products based
on SMOS and SMAP are derived by a slightly different approach, which assimilate the observed
brightness temperatures instead of the retrieved surface soil moisture products (Lannoy and Reichle,
2016). The assimilation of satellite-based soil moisture products in terrestrial carbon cycle models
has been described above.



### 3.3.5 Biomass

Continental-scale biomass maps have been produced from space using both radar and lidar; these
rely on the returns from transmitted power, so are known as active sensors. Biomass here refers to
above-ground biomass (AGB), since there are no methods to measure the below-ground component,
and this is typically inferred from AGB using allometric equations. Furthermore, the emphasis is on
the AGB of forests, although a global dataset of AGB in all biomes for the period 1993-2012 has
been produced based on global passive microwave satellite data, hence with spatial resolution of 10
km or coarser (Liu et al., 2015).

Using long time series of C-band radar data provided by the ESA Envisat satellite, the growing
stock volume of northern hemisphere boreal and temperate forests has been estimated (Santoro et al.,
2011). Although available at 0.01° resolution, the accuracy of growing stock volume at this scale
is comparatively poor, and spatial averaging provides more reliable results: at 0.5° spacing, esti-
mated growing stock volume has a relative accuracy of 20-30% when tested against inventory data
(Santoro et al., 2013). Thurner et al. (2014) used this product to derive the carbon stock (above- and
below-ground) in boreal, temperate mixed and broadleaf, and temperate coniferous forests of forests
above 30° N (40.7, 24.5 and 14.5 PgC respectively). These values have estimated accuracies of
around 33-39% under a conservative approach to estimate uncertainty.

For tropical forests, the key sensor is the Geoscience Laser Altimeter System (GLAS) onboard
the Ice, Cloud and land Elevation Satellite (ICESat) which failed in 2009 (Lefsky, 2010). Its archive
of forest height estimates was the core dataset exploited to produce two pan-tropical biomass maps
(Saatchi et al., 2011; Baccini et al., 2012) at grid scales of 1 km and 500 m respectively; Saatchi et al.
(2011) also provide a map of the errors associated with the biomass estimates at each pixel. This is
produced by combining measurement errors, allometry errors, sampling errors, and prediction errors,
which are treated as independent and spatially uncorrelated. Further details are given in the supple-
mentary material to Saatchi et al. (2011). In an attempt to resolve differences between these two
maps, Avitabile et al. (2016) used an independent reference dataset of field observations to remove
the biases in the maps and then combined them to estimate the AGB in the tropical belt (23.4° S to
23.4° N). Testing against a reference dataset not used in the fusion process indicated that the fused
map had a RMSE 15-21% lower than that of the input maps and nearly unbiased estimates.

However, there are unresolved questions about large-scale biomass patterns across the Amazon
inferred from in situ and satellite data. Biomass maps derived from satellite data in Saatchi et al.
(2011) and Baccini et al. (2012) differ significantly from each other and from biomass maps derived
from in situ plots distributed across Amazonia using kriging (Mitchard et al., 2014). Neither satel-
lite product exhibits the strong increase in biomass from southwestern to northeastern Amazonia
inferred from in situ data. Mitchard et al. (2014) attributed this to failure to account for gradients in
wood density and regionally varying tree height-diameter relations when estimating biomass from
the satellite data. Saatchi et al. (2015) reject this analysis and claim that the trends and patterns in



Mitchard et al. (2014) are erroneous and a consequence of inadequate sampling. Resolving this disagreement is of fundamental importance since it raises basic questions about accuracy, uncertainty, and representativeness for both in situ and satellite-derived biomass data.

The next 4-5 years will dramatically improve our global knowledge of biomass, with the launch of three missions aimed at measuring forest structure and biomass. The ESA BIOMASS mission

(European Space Agency, 2012), to be launched in 2021, is a P-band radar that will provide near-global measurements of forest biomass and height. Around the same time the NASA-ISRO SAR mission (NISAR) based on an L-band sensor will be deployed, providing measurements of biomass in lower biomass forests (up to 100 t ha$^{-1}$). These highly complementary missions will be further complemented by the NASA Global Ecosystem Dynamics Investigation vegetation lidar to be placed

on the International Space Station around 2019; this aims to provide the first global, high-resolution observations of the vertical structure of tropical and temperate forests, from which biomass may be estimated.

## 4  Conclusions

In the context of carbon cycle data assimilation this paper reviews the requirements and summarises

the availability and characteristics of some selected observations with a special focus on remotely sensed Earth observation data. The paper also briefly recapitulates the assimilation systems capable of integrating these data, a more comprehensive description of the underlying formalism is given in Rayner et al. (2016) while MacBean et al. (2016) discuss the implementation strategies for a multiple data assimilation system and its impacts on the results. To take maximum advantage of these

data streams in carbon cycle data assimilation studies it is of utmost importance to have the appropriate knowledge of the observational characteristics of the observational data, here with a focus on atellite products. This includes an understanding of the observable and its representativeness in order to develop the appropriate observation operator (see also Kaminski and Mathieu, 2016) but also the structure of any biases, random errors and error covariances (that is both the diagonal and

off-diagonal elements quantifying the correlations between different observations).

The benefit of using multiple data streams in a CCDAS lies in the complementarity of the data, and thus in the ability to constrain different components of the underlying process model. For example, while FAPAR data constrain mainly the phenology component of a terrestrial carbon cycle model, soil moisture data, in contrast, constrain the hydrological component, but both components

are important elements of the model and determine the simulated carbon fluxes. In fact, because of the model internal interactions and feedbacks among the components the simultaneous assimilation of complementary observations has synergistic effects such that the constraint is larger than the sum of the individual constraints as shown for instance by Kato et al. (2013) assimilating observations of FAPAR and latent heat flux.



As a final remark one important aspect of observational data is their continuity since much of the important information is contained in response to climate anomalies. Fortunately, the set up of operational observeing systems such as ICOS for in-situ data or Copernicus for satellite data has created the necessary infrastructure to ensure such a long-term perspective in the provision of Earth observations.



**Appendix A: List of Acronyms**

| | |
|---|---|
| ACE-FTS | Atmospheric Chemistry Experiment - Fourier Transform Spectrometer |
| AGB | Above Ground Biomass |
| AIRS | Atmospheric Infrared Sounder |
| AMSR2 | Advanced Microwave Scanning Radiometer 2 |
| AMSR-E | Advanced Microwave Scanning Radiometer - Earth Observing System |
| ASCAT | Advanced Scatterometer |
| ATSR | Along Track Scanning Radiometers |
| AVHRR | Advanced Very High Resolution Radiometer |
| CCDAS | Carbon Cycle Data Assimilation System |
| CCI | Climate Change Initiative |
| ECMWF | European Centre for Medium-Range Weather Forecasts |
| ECV | Essential Climate Variable |
| EO | Earth Observation (in this form generally understood as from space) |
| ESA | European Space Agency |
| FAPAR | Fraction of Absorbed Photosynthetically Active Radiation |
| FLEX | FLuorescence EXplorer |
| GCOM-W1 | Global Change Observation Mission 1st-Water |
| GLAS | Geoscience Laser Altimeter System |
| GLEAM | Global Land Evaporation Amsterdam Model |
| GOME-2 | Global Ozone Monitoring Experiment-2 |
| GOSAT | Greenhouse Gases Observing Satellite |
| GPP | Gross Primary Productivity |
| IASI | Infrared Atmospheric Sounding Interferometer |
| ICOS | Integrated Carbon Observing System |
| ICESat | Ice, Cloud and land Elevation Satellite |
| ISRO | Indian Space Research Organisation |
| JAXA | Japan Aerospace Exploration Agency |
| JRC-TIP | Joint Research Centre – Two-stream Inversion Package |
| MERIS | Medium Resolution Imaging Spectrometer |
| MIPAS | Michelson Interferometer for Passive Atmospheric Sounding |
| MISR | Multiangle Imaging SpectroRadiometer |
| MODIS | Moderate Resolution Imaging Spectroradiometer |
| NASA | National Aeronautics and Space Administration |
| NDVI | Normalized Difference Vegetation Index |
| NIR | Near Infrared |
| Obs4Mips | Observations for Model Intercomparisons Project |
| OCO-2 | Orbiting Carbon Observatory 2 |
| OE | Optimal Estimation |





| | |
|---|---|
| PDF | Probability Density Function |
| SAR | Synthetic Aperture Radar |
| SCIAMACHY | Scanning Imaging Absorption Spectrometer for Atmospheric Chartography |
| SeaWiFS | Sea-viewing Wide Field-of-view Sensor |
| SEVIRI | Spinning Enhanced Visible and InfraRed Imager |
| SIF | Sun-Induced Fluorescence |
| SMAP | Soil Moisture Active Passive |
| SMOS | Soil Moisture Ocean Salinity |
| SWIR | Shortwave Infrared |
| TANSO-FTS | Thermal And Near infrared Sensor for carbon Observations - Fourier Transform Spectrometer |
| TCA | Triple Collocation Analysis |
| TCCON | Total Carbon Column Observing Network |
| TCOS | Terrestrial Carbon Observation System |
| TROPOMI | TROPOspheric Monitoring Instrument |

*Acknowledgements.* M.B. has received funding from ESA via the GHG-CCI project. W.D. is supported by the "TU Wien Wissenschaftspreis 2015" a personal grant awarded by the Vienna University of Technology. Fig. 4 was kindly provided by Philipp Köhler, California Institute of Technology. We acknowledge the support from the International Space Science Institute (ISSI). This publication is an outcome of the ISSI's Working Group

on "Carbon Cycle Data Assimilation: How to consistently assimilate multiple data streams".



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




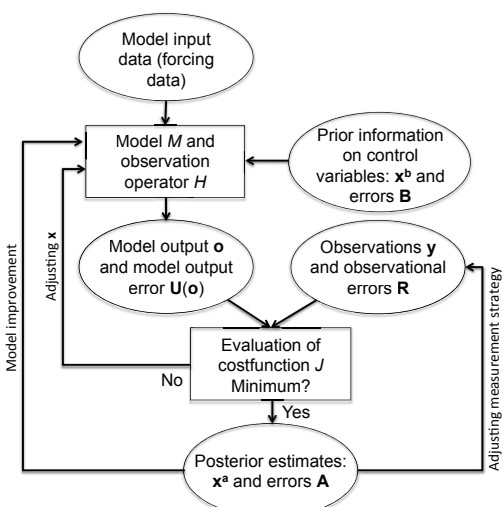

**Figure 1.** Schematic of a data assimilation system with **x** being the control vector containing the quantities to be updated by the assimilation. The inner loop ('Model-data comparison' box to 'Model and observation operator' box) indicates the assimilation process. Often, the analysis of residuals in model data comparison lead to either model improvements or adjustment of the measurement strategies (outer loops).

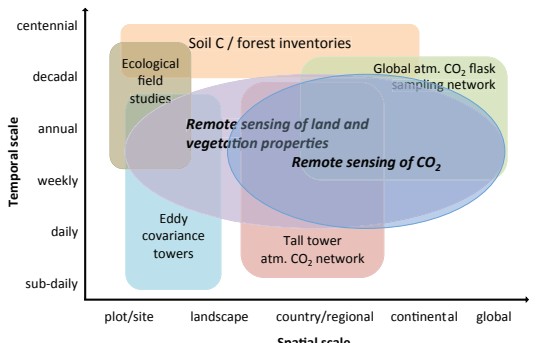

**Figure 2.** Space-time diagramme for a range of observations relevant for a Terrestrial Carbon Observation System.

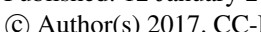


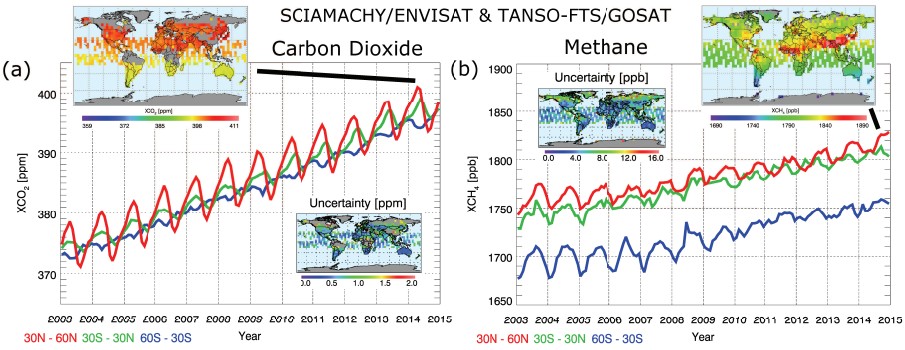

**Figure 3.** Timeseries of satellite-derived XCO2 in 3 latitude bands (see annotation bottom left, e.g. red line: 30o-60oN) and maps showing the spatial distribution of XCO2 for April 2014 (top left map) and corresponding XCO2 uncertainty (bottom). (b) As (a) but for XCH4 (maps: September 2014).

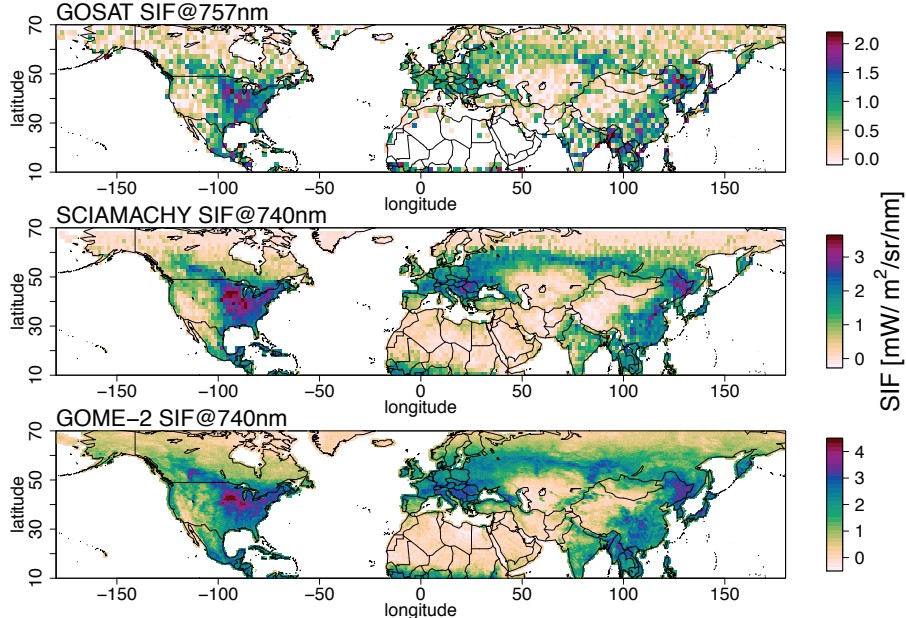

**Figure 4.** Maps of sun-induced fluorescence (SIF) for July 2010 derived from GOSAT, GOME-2 and SCIA-MACHY satellite data.




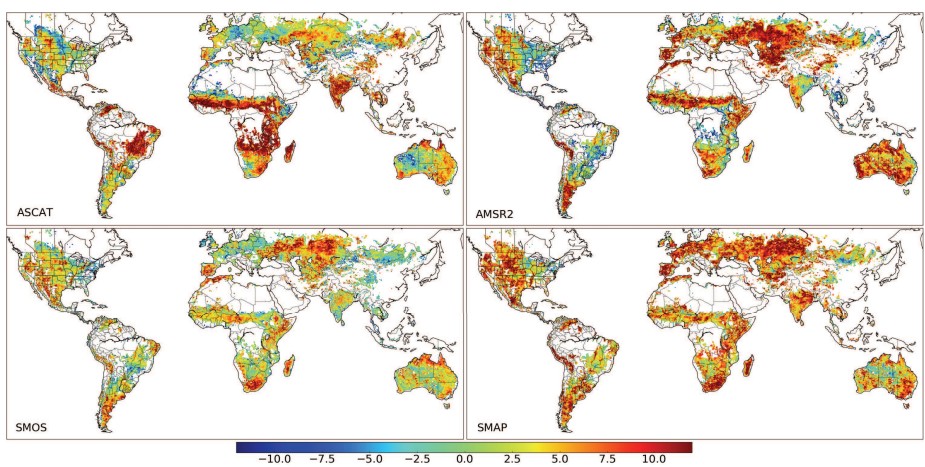

**Figure 5.** Signal-to-noise ratio (in dB), estimated with the Triple Collocation Analysis for four different satellite-based soil moisture products and a Land Surface Model. a) MetOp-A ASCAT based on the TU Wien method (Wagner et al., 1999); b) AMSR2 based on the LPRM model (Owe et al., 2008); c) SMOS L3 (Kerr et al., 2010); d) SMAP (Jackson, 1993). An SNR of -3 indicates a signal variance that is half of the noise variance, an SNR of 0 a signal variance equal to the noise variance, an SNR of 3 a signal variance that is twice the noise variance, and so on. In areas without data the TC could not be computed, e.g. because of too few observations in one of the datasets. For details see Gruber et al. (2016b).





**Table 1.** Overview SCIAMACHY/ENVISAT and TANSO-FTS/GOSAT XCO2 and XCH4 Level 2 data products (individual ground-pixel retrievals). For some products also Level 3, i.e. gridded data products are available (e.g. for CO2_SCI_WFMD and CH4_SCI_WFMD from http://www.iup.unibremen.de/sciamachy/NIR_NADIR_WFM_DOAS/ and merged SCIAMACHY and TANSO-FTS XCO2 and XCH4 products in Obs4MIPs format from http://www.esa-ghg-cci.org/)

| Parameter | Sensor | Available at: Product (Reference) |
|---|---|---|
| XCO2 | SCIAMACHY | http://www.esa-ghg-cci.org/ |
| | | CO2_SCI_BESD (Reuter et al., 2011) |
| | | CH4_SCI_WFMD (Schneising et al., 2011) |
| | TANSO | http://www.gosat.nies.go.jp/en/ |
| | | NIES operational GOSAT (Yoshida et al., 2013) |
| | | http://www.esa-ghg-cci.org/ |
| | | CO2_GOS_OCFP (Cogan et al., 2012) |
| | | CO2_GOS_SRFP/RemoTeC (Butz et al., 2011) |
| | | http://www.iup.uni-bremen.de/ heymann/besd_gosat.php |
| | | GOSAT BESD (Heymann et al., 2015) |
| | | http://disc.sci.gsfc.nasa.gov/acdisc/documentation/ACOS.html |
| | | NASA ACOS (Crisp et al., 2012) |
| | SCIAMACHY & TANSO merged | http://www.esa-ghg-cci.org/ |
| | | CO2_EMMA (Reuter et al., 2013) |
| | OCO-2 | http://disc.sci.gsfc.nasa.gov/OCO-2 |
| | | NASA OCO-2 (Boesch et al., 2011) |
| XCH4 | SCIAMACHY | http://www.esa-ghg-cci.org/ |
| | | CH4_SCI_WFMD (Schneising et al., 2011) |
| | | CH4_SCI_IMAP (Frankenberg et al., 2011a) |
| | TANSO | http://www.gosat.nies.go.jp/en/ |
| | | NIES operational GOSAT (Yoshida et al., 2013) |
| | | http://www.esa-ghg-cci.org/ |
| | | CH4_GOS_OCPR (Parker et al., 2011) |
| | | CH4_GOS_SRPR/RemoTeC (Butz et al., 2010) |
| | | CH4_GOS_OCFP (Parker et al., 2011) |
| | | CH4_GOS_SRFP/RemoTeC (Butz et al., 2011) |
| | SCIAMCHY & TANSO merged | http://www.esa-ghg-cci.org/ |
| | | CH4_EMMA (Reuter et al., 2013) |



**Table 2.** Characteristics of a variety of FAPAR products, more details and products are provided by
D'Odorico et al. (e.g. 2014); Pickett-Heaps et al. (e.g. 2014).

| Name | Time period | Temporal resolution | Definition | Reference |
|------|------|------|------|------|
| MODIS | 2000-present | 8 days | Green canopy, direct radiation | Myneni et al. (2002) |
| SeaWiFS[1] | 1997-2006 | 10 days | Green canopy, diffuse radiation | Gobron et al. (2006) |
| TIP-MODIS | 2000-present | 16 days | FAPAR/Green canopy, diffuse radiation | Pinty et al. (2011b) |
| TIP-GlobAlbedo | 2002-2011 | 8 days | FAPAR/Green canopy, diffuse radiation | Disney et al. (2016) |
| Vegetation | 1999-present | 10 days | FAPAR, direct radiation | Baret et al. (2007) |

1 The same algorithm is also used for MERIS, spanning a period form 2003-2012 with a 1 km, 10 day resolution.

**Table 3.** Selected characteristics of operating and planned spaceborne instruments able to deliver SIF data.
Names of upcoming instruments are highlighted in italics. NIR stands for near-infrared. It must be noted that
GOME-2 on MetOp-A has been operating with a reduced pixel size of $40\times40$ km$^2$ since July 2013.

| | Time period | Overpass time | Spectral sampling | Global coverage | Spatial resolution | Temporal resolution |
|------|------|------|------|------|------|------|
| GOSAT | 2009–today | Midday | NIR | No | 10 km diam. | 3 days |
| GOME-2 | 2007–today | Morning | red & NIR | Yes | $40\times80$ km$^2$ | <2 days |
| SCIAMACHY | 2003–2012 | Morning | red & NIR | Yes | $30\times240$ km$^2$ | <3 days |
| OCO-2 | 2014–today | Midday | NIR | No | $1.3\times2.3$ km$^2$ | 16 days |
| *TROPOMI* | ~2017 | Midday | red & NIR | Yes | $7\times7$ km$^2$ | <1 day |
| *FLEX* | ~2022 | Morning | red & NIR | Yes | $0.3\times0.3$ km$^2$ | <27 days |



**Table 4.** Current (pre-)operational global soil moisture missions and products (for abbreviations / acronyms see List of Acronyms

| Mission | Organisation | Measurement concept | Band | Mission start | Data access |
|---|---|---|---|---|---|
| MetOp - ASCAT | EUMETSAT | Real aperture radar (scatterometer) | C-band | Jan. 2007 | http://hsaf.meteoam.it/soil-moisture.php http://land.copernicus.eu/global/products/swi |
| SMOS | ESA | Interferometric radiometer | L-band | Nov. 2009 | http://www.catds.fr/ |
| GCOM-W1 AMSR2 | JAXA | Radiometer | C-band | May 2012 | http://www.vandersat.com/ http://suzaku.eorc.jaxa.jp/GCOM_W/ |
| SMAP | NASA | Radiometer & radar[1] | L-band | Jan. 2015 | http://smap.jpl.nasa.gov/ |
| Sentinel-1 | ESA/ Copernicus | Synthetic aperture radar | C-band | Apr. 2014 | https://www.eodc.eu/ |
| CCI | ESA | Combined scatterometer and radiometer | L-, C-, X- Ku-band | Nov. 1978 | http://www.esa-soilmoisture-cci.org |

1 SMAP's radar failed in July 2015