# Peer review of "Reviews and syntheses: Systematic Earth observations for use in terrestrial carbon cycle data assimilation systems"

_Biogeosciences, 2016_

## Referee Comment (RC1) · N. MacBean (Referee) · 16 Feb 2017

Review of Scholze et al. "Reviews and syntheses: Systematic Earth observations for use in terrestrial carbon cycle data assimilation systems"

Scholze et al. present a rigorous, in-depth review of the major observation types that are used to constrain the carbon cycle and related variables in terrestrial biosphere models. Such a synthesis is timely given the increasing availability of a variety of observations, as well as the pressing need to reduce uncertainty in model simulations of the current and future carbon budget. The strength of the paper lies in the extensive description of the main types of EO observations that can be used in a carbon cycle data assimilation (DA) system. As such I would strongly recommend this paper to any

colleagues wishing to use these data for model optimization (and benchmarking). I have made some comments below that I think would further improve the structure and clarity of the manuscript, as well as suggesting some extra references – although I appreciate the authors are not aiming to provide an exhaustive review.

General comments

Firstly, I appreciate the distinction the authors try to make between their review and that of Raupach et al. (2005) and Ciais et al. (2014) by placing an emphasis on EO data versus in situ atmospheric $CO_2$ and eddy covariance data; however, these two sources of data are one of the most widely used in carbon cycle DA studies, and therefore I think it is worth having a separate section that briefly summarizes these data and their uncertainties, while keeping the focus on EO. Otherwise the description of updates to the eddy covariance uncertainty estimates in the general section on observational uncertainties (Section 3.1) could feel out of place. In addition, Section 3.2 discusses operational carbon observing systems which currently include many in situ networks.

Secondly, I suggest a slight re-structuring so all the examples of DA studies with these data are incorporated into one specific section, and possibly after the description of the different types of observations. Currently, there are examples in Section 2.3 and the introduction to 3.3. Whilst the examples given in the latter are specifically pertinent to EO data, the use of EO data in an assimilation system has been discussed already in Section 2.3, and therefore the lines are somewhat blurred.

Finally, it would be good to include websites/references for data access in all data tables (as in Table 1), and, given the important emphasis on observation uncertainty, note if uncertainty estimates come with the data.

Specific comments

Section 2.1

Lines 109-111: worth pointing out that a better fit between the posterior maximum

likelihood simulation and the observations does not necessarily mean you have the correct parameters and/or model structure (e.g. MacBean et al., 2016)

Section 2.2: The distinction between sequential and variational DA could be slightly confusing for the lay reader. I suggest the following:

Line 133: make it clear that sequential assimilation happens at the point of having an observation – otherwise one may wonder "at which discrete time steps?"

Lines 137-139: I think this could read as if J is only evaluated in the variational approach (though that may be helped by changing the caption of Figure 1 – see below). I suggest that instead of just discussing the inner loop you could make a distinction about when J is evaluated and at what point the minimum is found for both approaches. In addition, it might be helpful to the reader to have a sentence that qualitatively describes what the cost function represents and to explicitly say that the aim is to minimize the cost function around lines ~132-139.

Figure 1: I like this figure, but I cannot see a "Model-data comparison" box as you describe in the caption. I guess you mean "Evaluation of J"?

Section 2.3: Line 195: Maybe add paper by Bloom and Williams (2015) and latest CLM paper by Post et al. (2017)? Line 200: maybe worth adding "...same cost function value at the minimum"? Line 203: Could add Thum et al. (2017) here

Section 3.1: Worth mentioning that observation errors in a DA system should include the models errors, and what could give rise to errors in the model?

Section 3.2: This section is very focused on Europe. It would be worthwhile detailing efforts that are underway in other regions, e.g. example the NASA Carbon Monitoring System (http://carbon.nasa.gov). This section also feels a little out of place. I might suggest incorporating it into the introduction to Section 3 or having it as a perspectives section at the end of the article.

Section 3.3: Lines 306-308: worth pointing out that using level 2 products may increase the observation uncertainty, particularly given parameters/processes implemented in retrieval algorithm may not be consistent with corresponding equivalent parameters/processes in the underlying model (and that this may be a benefit of using level 1 products – e.g. Quaife et al., 2008). Also perhaps worth explaining that for vegetation activity that VIs are an intermediate step in that they are "lower order" products – i.e. they are raw radiances but also do not require a complex retrieval algorithm; instead they require an atmospheric transport model and limited calculations

Line 316-318: worth including that NDVI has also been used (e.g. MacBean et al., 2015a), and the advantages/disadvantages of using VIs

Line 318: Forkel et al. (2015) is another example of the use of FAPAR with a terrestrial model.

Line 321: and by optimizing parameters related to phenology and photosynthesis (MacBean et al., 2015b)

Line 322: Saying "Also assimilation of XCO2" comes a bit out of the blue here as you have just been talking about vegetation activity. Please could you say what is meant by XCO2, or refer to section 3.3.1.

Lines 325-332: Other examples of the impact of soil moisture (and LAI and FAPAR) data assimilation on LAI and C fluxes include the work at CNRM with the ISBA-A-gs model, e.g. Barbu et al. (2014).

Line 334: several studies have demonstrated the added benefit of aboveground biomass, including articles already cited (Richardson et al., 2010; Williams et al., 2005; Keenan et al., 2012). Might be worth listing a few examples, or, combining this section with aforementioned examples of C cycle related DA studies (section 2.3).

Line 340: LAI has been used in C cycle DA (see Barbu et al., 2014). Further to my comment on VIs above, perhaps it would be worth explaining somewhere in the text the differences between using VIs, FAPAR and LAI, why one would use one vs another?

Line 341: Worth mentioning the dataset of Li et al. (2011) that has been used in several studies investigating trends in biomass. In fact, I expect that VOD data will be increasingly widely used for optimizing biomass in terrestrial biosphere models, and therefore I would suggest adding a discussion of what these data actually represent in Section 3.3.5 (i.e. how reliably can you estimate biomass (leaf or total aboveground?) from what is essentially a measure of water content).

Line 343: I think that LST might be used in a similar manner to soil moisture in DA in the future, and not just as an input/boundary condition. Therefore perhaps it can be included with VOD in this context?

Section 3.3.2 Lines 477: you mean significant difference in the absolute magnitude between the products (as the temporal and spatial patterns are quite consistent, as you state)? This was also a conclusion drawn by D'Odorico et al. (2014) and, to some extent, Tao et al. (2015); therefore, it is worth mentioning that these studies agree on this point.

As mentioned above, here or elsewhere I think it would be beneficial to have a discussion of the use of VIs and LAI. Arguably LAI is the variable that is most closely linked to standard terrestrial model state variables, therefore the reader should understand why one might use any of these three options for optimizing vegetation dynamics/activity, and the advantages/disadvantages of each. For example, if a modeler is mostly concerned with optimizing the overall magnitude of vegetation activity, careful choice of which FAPAR product to use is important (likely the same for LAI). If they are more concerned with temporal dynamics, one could argue that using a normalized lower order product (e.g. NDVI) that does not require such a sophisticated retrieval algorithm might be more appropriate. Perhaps you do not agree! But in any case, a discussion would be useful here.

At the end of this section there is a particular focus on the JRC-TIP FAPAR product as opposed to one of the others, MODIS for example. It would be good to explain

the reason for this choice, or to see more information on some of the other commonly available products.

Lines 484-485: Please could you be clearer how the products in this sentence link to Table 2? JRC MGVI is not described in Table 2 for example.

Section 3.3.4 Lines 489-491: Although I appreciate you do not wish to provide an exhaustive description of retrieval algorithms, I think it would be helpful to qualitatively describe the difference between passive and active retrieval algorithms in one or two sentences here, as well as the fact different algorithms may produce either volumetric water content (absolute values) vs relative soil moisture values.

I would be interested to see a discussion of GRACE land water content in this section.

Section 3.3.5 Line 670: do not need to reiterate what an active sensor is here.

Line 675: I see you do refer to the VOD product of Liu et al. here. Still, I think it would be beneficial to detail that this is based on VOD data and describe briefly how VOD are derived (following on from the mention of VOD in the soil moisture section) and how biomass is estimated from VOD and their expected use/value for optimizing biomass (as discussed above), as well as for better understanding discrepancies in other sources of biomass data that you discuss towards the end of Section 3.3.5.

Lines 676-684: updated reference: Santoro et al. (2015) – update to aforementioned papers providing biomass estimates across a wider range of biomes in the northern hemisphere.

Line 711: Could you provide the biomass limit that the P-band BIOMASS mission will be able to resolve (to compare with the NISAR mission)?

Given you mention the international soil moisture network in Section 3.3.4, it may be worth mentioning the international tree ring data bank in this section (https://data.noaa.gov/dataset/international-tree-ring-data-bank-itrdb), as these data represent a promising new direction for optimizing biomass across a range of biomes.

Minor comments and typos Line 252: maybe "between" better than "among"? Line 309-310: sentence could be simplified Line 111: benchmark Line 135: measurement Line 145: knowledge Line 234 and 241: related Line 255: diagonal Line 311: terrestrial Line 315: biogeochemical Line 420: that than Line 440: reflectance Line 576: complementarily Line 727: satellite Line 1395: Updated Thum et al. (2016) reference – see below.

References:

Barbu, A. L., Calvet, J.-C., Mahfouf, J.-F., and Lafont, S.: Integrating ASCAT surface soil moisture and GEOV1 leaf area index into the SURFEX modelling platform: a land data assimilation application over France, Hydrol. Earth Syst. Sci., 18, 173-192, doi:10.5194/hess-18-173-2014, 2014.

Bloom, A. A. and Williams, M.: Constraining ecosystem carbon dynamics in a data-limited world: integrating ecological "common sense" in a model–data fusion framework, Biogeosciences, 12, 1299-1315, doi:10.5194/bg-12-1299-2015, 2015.

Forkel, M., Carvalhais, N., Schaphoff, S., v. Bloh, W., Migliavacca, M., Thurner, M., and Thonicke, K.: Identifying environmental controls on vegetation greenness phenology through model–data integration, Biogeosciences, 11, 7025-7050, doi:10.5194/bg-11-7025-2014, 2014.

Liu, Y. Y., R. A. M. de Jeu, M. F. McCabe, J. P. Evans, and A. I. J. M. van Dijkâ(2011), Global long-term passive microwave satellite-based retrievals of vegetation optical depth,âGeophys. Res. Lett., 38, L18402, doi:10.1029/2011GL048684.

MacBean, N., Maignan, F., Peylin, P., Bacour, C., Bréon, F.-M., and Ciais, P.: Using satellite data to improve the leaf phenology of a global terrestrial biosphere model, Biogeosciences, 12, 7185-7208, doi:10.5194/bg-12-7185-2015, 2015a.

MacBean, N., F Maignan, P Lewis, L Guanter, P Koehler, C Bacour, P Peylin, J Gomez-Dans, M Disney, F Chevallier, A Model-Data Fusion Approach for Constraining Modeled GPP at Global Scales Using GOME2 SIF Data, Abstract #B43H-0649 presented at 2015, Fall Meeting, AGU, San Francisco, CA, 14-18 December, 2015b http://adsabs.harvard.edu/abs/2015AGUFM.B43H0649M

Post, H., J. A. Vrugt, A. Fox, H. Vereecken, and H.-J. H. Franssen (2017), Estimation of Community Land Model parameters for an improved assessment of net carbon fluxes at European sites, J. Geophys. Res. Biogeosci., 122, doi:10.1002/2015JG003297.

Quaife, T., Lewis, P., De Kauwe, M., Williams, M., Law, B.E., Disney, M. and Bowyer P. (2008) Assimilating canopy reflectance data into an ecosystem model with an Ensemble Kalman Filter. Remote Sensing of Environment, 112, 1347-1367

Santoro M, Beaudoin A, Beer C, Cartus O, Fransson JE, Hall RJ et al. (2015).ÂăForest growing stock volume of the northern hemisphere: Spatially explicit estimates for 2010 derived from Envisat ASAR. Remote Sensing of Environment, 168, 316-334.

Thum, T., N. MacBean, P. Peylin, C. Bacour, D. Santaren, B. Longdoz D. Loustau and P. Ciais (2017) The potential benefit of using forest biomass data in addition to carbon and water flux measurements to constrain ecosystem model parameters: case studies at two temperate forest sites, Agricultural and Forest Meteorology, 234, 48-65.

---

## Referee Comment (RC2) · T. Kaminski (Referee) · 12 Apr 2017

This manuscript presents several observational data streams that provide useful constraints in a Carbon Cycle Data Assimilation System (CCDAS). Such a contribution to the special issue covering the observational aspects, is very useful as it provides information that is complementary to the remainder of the articles in the special issue, which present application examples, overviews, or methodology. It is also reasonable to focus on a few Earth Observation (EO) data streams that were not covered in previous review articles on the subject. And it is reasonable not to focus only on direct observations of the carbon cycle but also address data streams such as soil moisture, which, through the process model, acts as an indirect constraint on the carbon cycle.

[Figure]

This special issue is one of the outputs of a working group of the international space science institute's working group "Carbon Cycle Data Assimilation: How to Consistently Assimilate Multiple Data Streams", the other one was a summer school, and one intention of this special issue is to provide further reading material to the students. In this context, the manuscript offers the right level of detail as a starting point, with many references to further material.

This is a good paper. My following comments are all minor and attempt

- to render the manuscript more useful for the target readers,

- to focus on the overall subject of the ISSI group and the special issue "consistent assimilation of multiple data streams", and

- to provide links to the other contributions.

The authors should address those they find useful.

1. Model-data fusion and data assimilation: L 37 states that both terms mean the same. Is this true? If yes, I suggest to keep one of the two for the rest of the manuscript instead of switching between the two. If not – maybe because by model-data fusion we could also understand some blending of observations with pre-computed model output – then be more precise in the definitions here and below use the appropriate term depending on context.

2. L 47: "new observation": maybe rather "new data stream" or "new type of observation"

3. L64: "In contrast to Ciais et al. (2014), who focus on carbon-cycle observations, we focus here on any kind of relevant observational data to be (potentially) assimilated in a terrestrial carbon cycle data assimilation system (CCDAS). In

a CCDAS the observations are used to constrain the underlying model (i.e. to move model output quantities closer to the observations and reduce their posterior uncertainties) usually by parameter optimisation." This formulation could be improved. "any kind of" could be dropped in the first sentence, and the second sentence could read (for example): In a CCDAS non-carbon observations can be exploited to constrain the simulated carbon cycle indirectly through the relations implemented in the process model. Such observational constraints act by ruling out combinations of the unknowns in a CCDAS (typically a combination of process parameters, initial- or boundary conditions) that are inconsistent with the observations and thereby reduce uncertainties in the simulated output."

4. L73: "Our focus lies on the terrestrial carbon cycle, because of the higher spatial and temporal variability in the net exchange fluxes and their associated higher uncertainties than form the ocean and anthropogenic components." Maybe not true on all relevant scales. Maybe just drop the sentence, no need to justify the terrestrial focus in this context.

5. L101: in fact the weighting is in inverse proportion of the uncertainty, also appears below where Eq 1 is described

6. L103: either is maybe not appropriate?

7. L118: Maybe you want to put: "Here, we follow the notation as introduced by Rayner et al. (2016)" at the beginning of the subsection, i.e. before you start using their notation.

8. As we are dealing with assimilation of "multiple data streams" you could mention that usually each data steam requires its own observation operators, and in fact already here refer to Kaminski and Mathieu (2016/7), maybe even their octopus Figure. And for Eq 1 you could say that, for convenience of notation, now you combine all of them into a single H().

9. L122: Maybe drop this sentence. In fact the state are mixing ratios.

10. L133: "thus evolves"?

11. L136: "optimality" maybe you can find a better word? Maybe "adequacy"?

12. Figure 1 is confusing in several respects (prior info enters cost, inner/outer loop to be confused with NWP terminology, U(o) not necessarily only a model output, cost function at minimum does not imply availability of A, etc ...). Maybe you just want to drop it with the two sentences that describe it?

13. L113-150 starting with "From Equation ..." could also be clearer, shortened or dropped (It does not follow from Eq. 1 that uncertainties are to be taken into account, but Eq. 1 follows from combining PDF descriptions of prior, observations, and model with a few simplifications, Mean and variance are not sufficient to characterise a multi-variate Gaussian, ...) Same holds for next paragraph ("assimilation problem is Gaussian" does not make sense; division by the matrix B is not straightforward...) Maybe just explain variables in Eq. 1 and then directly move to the paragraph starting with Rayner et al. (2016).

14. L161: The "and" between citations is missing ("citep" would have worked for multiple citations), same problem occurs a few times further down below.

15. L205: what about non-linear observation operators?

16. Section 3.1: It is good to introduce the different forms of errors. It would also be instructive to provide definitions of precision and accuracy.

17. L242: Is is worth mentioning that the scale at which we trust the model may be larger than a grid cell? L247: "In the case of satellite-based observations the representation error also includes errors in inferring a biophysical quantity from the photons measured at the sensor. We come back to this issue later." I would think that such errors in the retrieval rather go into into the above two categories?

18. L 255: "they affect the prediction of the optimal solution in the same way as" could maybe be replaced by "they have considerable impact on the solution. This is because of their influence on the weight of the respective observations in the cost function." It is very good you stress this point. In fact you should take it up in the presentation of each data stream. So far it is only addressed in the XCO2 and the biomass sections.

19. L 263: What is inhomogeneous variance?

20. Section 3.2: Maybe add reference/web page of ICOS? Is is worth mentioning similar programmes outside Europe? "The measurements are designed": maybe better "the network" or "the observing system"? L282: Paragraph may fit better into the beginning of section 3.3. Where you discuss the relevant observations provided by the sentinels, you are using our current perspecitve, i.e. S1-5. You could make this clear, because in a few years time are reader could wonder why you don't mention observations by S6 ... etc...

21. L 305 and 310: On L305 you write EO, then Earth Observation, then EO... something to be checked throughout ...

22. L344: For example Luke (2011) assimilates LAI.

23. Section 3.3.1: Is it worth to briefly explain how a total column value can be sensitive or insensitive to surface fluxes? L400: ")" should be "(" L413: You could mention how the aggregation of errors to the 5 degree grid was performed. L435: Maybe update reference to latest version of the CCI CAR. L439: Is is worth mentioning planned XCO2 missions?

24. Section 3.3.2: L445: remove one ")". L458: "closely follows the state of the vegetation" could be "is determined by the state of the canopy-soil system"L475: Disney et al. (2016) also compare two products. L 480: To simplify the sentence,

maybe move the part in brackets up to the definition of L460. L503: "see 2" should be "see table 2". Regarding correlation of uncertainty you might add on L506: after "periods.": To reduce disk space, by default, JRC-TIP products are delivered without correlations among the uncertainties between individual variables, even though these correlations are available. An estimate of uncertainty correlation in space and time is not provided. The JRC-TIP products derived from MODIS (collection 5) broadband albedos minimise temporal uncertainty correlation as each collection 5 albedo value is derived as integral exclusively of observations over non-overlapping 16-day periods.

25. Section 3.3.3: L 514: "directly related" In the context of data assimilation, is it worth mentioning that there are complex processes which require complex models as observation operators for SIF, in order to extract the maximal benefit from this data steam? L 524: "lies" could be "relies"? L 530: Why is the simplicity of the forward model related to the fact that least squares is applied, which might also work with complex models? L 540: Does "a compromise" make sense here? Isn't it rather the definition of the grid size that is determined by a compromise and the number of retrievals then just a function of this choice of grid size (plus the other factors mentioned)? When discussing the spatial and temporal sampling, it might be instructive to mention the variability in time as you do it in space.

26. Section 3.3.4: L600: "cost of the": maybe better "cost of long" L652: ",5." L658: "to improve the model's hydrology" in fact in a CCDAS we are after the indirect constraint on carbon, so this restriction may not be needed here?

27. Conclusions: L724: "observational characteristics of the observational data" maybe you meant error or uncertainty characteristics? L730: "correlations": "uncertainty correlations" or "error correlations" L732: "For example, while FA-PAR data constrain mainly the phenology component of a terrestrial carbon cycle model, soil moisture data, in contrast, constrain the hydrological component," see

above regarding the indirect constraints. You probably do not need to write this. In fact FAPAR can provide an important constraint on hydrology (see Kaminski et al., 2012)

28. Fig 3: I'd suggest to go for a 6 panel figure, the four maps are tiny; Better use degree symbol in capton.

29. Table 1: I suggest to replace "parameter" by "variable"

30. Table 4: You could add wave lengths to the bands, for many colleagues the band names don't mean anything.

There are quite a number of typos. Many of them (e.g. "Reflectamce-based" or "assessemt" or "observeing") can be detected by a spell checker...

---

## Author Comment (AC1) · 26 May 2017

Response to reviewer comments to manuscript "Reviews and syntheses:
Systematic Earth observations for use in
terrestrial carbon cycle data assimilation systems"

We thank the two reviewers for their careful inspection of the manuscript. In the following we address their comments point-by-point. We use text in italics to repeat the reviewer comments, normal text for our response, and bold faced text for quotations from the manuscript, with added text marked in colour.

**Comments by referee 1, Natasha MacBean:**

*Firstly, I appreciate the distinction the authors try to make between their review and that of Raupach et al. (2005) and Ciais et al. (2014) by placing an emphasis on EO data versus in situ atmospheric CO2 and eddy covariance data; however, these two sources of data are one of the most widely used in carbon cycle DA studies, and therefore I think it is worth having a separate section that briefly summarizes these data and their uncertainties, while keeping the focus on EO. Otherwise the description of updates to the eddy covariance uncertainty estimates in the general section on observational uncertainties (Section 3.1) could feel out of place. In addition, Section 3.2 discusses operational carbon observing systems which currently include many in situ networks.*

We have extended the description of the in situ atmospheric $CO_2$ and eddy covariance data in the beginning of Section 3 such that the update to the eddy covariance uncertainty estimates does not feel out of place. However, we do not include a whole new section on those data as it has extensively been covered elsewhere. Also, we have removed Section 3.2 and included a shortened version of the text in the Conclusions with reference to other international observation networks.

[revised manuscript text omitted]

*Secondly, I suggest a slight re-structuring so all the examples of DA studies with these data are incorporated into one specific section, and possibly after the description of the different types of observations. Currently, there are examples in Section 2.3 and the introduction to 3.3. Whilst the examples given in the latter are specifically pertinent to EO data, the use of EO data in an assimilation system has been discussed already in Section 2.3, and therefore the lines are somewhat blurred.*

As suggested, we have slightly restructured the DA examples in the manuscript, but we have kept them in two places (Sections 2.3 and 3.2) to distinguish between general examples (Sec 2.3) and examples making use of the EO data on which this manuscript focusses (Sec. 3.2). We have also clarified this approach in the manuscript.

Section 2.3
**Recent advances focus on multiple independent data stream assimilation to provide a more rigorous constraint on the multiple components of terrestrial ecosystem models and avoid equifinality, i.e. different parameter solutions providing the same cost function value at the minimum. Examples for such studies on local/regional scale are the assimilation of eddy covariance $CO_2$ fluxes together with observations of vegetation structural information or carbon stocks (e.g. Richardson et al., 2010; Keenan et al., 2012; Thum et al., 2017). The assimilation of multiple data streams can be performed either in a step-wise (e.g. Peylin et al., 2016) or simultaneous approach (e.g. Kaminski et al., 2012); in the case of non-linear models or non-linear observation operators only the simultaneous assimilation makes optimal use of the observations (MacBean et al., 2016).** **In Section 3.2 we provide more terrestrial carbon cycle data assimilation examples using some of the remotely sensed products discussed in the following.**

Section 3.2
**FAPAR has already been demonstrated to provide a strong constraint on terrestrial carbon and water fluxes through its impact on the phenology components of the carbon cycle model** **either by assimilating only FAPAR data (e.g. Knorr et al., 2010) or in combination with other data streams (e.g. Kaminski et al., 2012; Kato et al., 2013; Forkel et al., 2014).**

*Finally, it would be good to include websites/references for data access in all data tables (as in Table 1), and, given the important emphasis on observation uncertainty, note if uncertainty estimates come with the data.*

Except for Table 3 all other Tables contain websites or references to the data products. Since SIF is a relatively new product data access is distributed among several websites (not official ones in some case), which may not be maintained after some years. Therefore, we did not include URLs but nevertheless added example references to Table 3.

*Lines 109-111: worth pointing out that a better fit between the posterior maximum likelihood simulation and the observations does not necessarily mean you have the correct parameters and/or model structure (e.g. MacBean et al., 2016).*

Included this point as suggested:

**In contrast, data assimilation, in particular when used for parameter optimisation, potentially identifies structural model and/or data deficiencies if the model-data mismatch (or the benchmark test) is still inadequate after optimisation (see also Figure (1)).** **On the other hand, a better fit between the posterior maximum likelihood simulation (i.e. using the optimised parameters) and the observations is not necessarily an indication for correct parameters and/or model structure as has been pointed out by MacBean et al. (2016).**

*Section 2.2: The distinction between sequential and variational DA could be slightly confusing for the lay reader. I suggest the following:*
*Line 133: make it clear that sequential assimilation happens at the point of having an observation – otherwise one may wonder "at which discrete time steps?"*

To make the distinction between sequential and variational DA clearer, we changed this sentence to:
**We distinguish two basic approaches in data assimilation: sequential assimilation, which assimilates observations subsequently at discrete model time-steps, and variational assimilation,…**

*Lines 137-139: I think this could read as if J is only evaluated in the variational approach (though that may be helped by changing the caption of Figure 1 – see below). I suggest that instead of just discussing the inner loop you could make a distinction about when J is evaluated and at what point the minimum is found for both approaches. In addition, it might be helpful to the reader to have a sentence that qualitatively describes what the cost function represents and to explicitly say that the aim is to minimize the cost function around lines 132-139.*

Changed the wording to:
**In the sequential approach the assimilation loop is evaluated sequentially over time following the dynamics of the model. In the case of variational assimilation the assimilation loop is evaluated iteratively (assuming a non-linear model). Both cases evaluate a cost function J, formulated in the Bayesian framework as:**

*Figure 1: I like this figure, but I cannot see a "Model-data comparison" box as you describe in the caption. I guess you mean "Evaluation of J"?*

Indeed, corrected. We have also slightly updated the figure and caption following the suggestion by referee 2.
**The loop between the 'Evaluation of J' box to 'Model and observation operator' box) indicates the assimilation process (assimilation loop). Often, the analysis of residuals in model-data comparison leads to either model improvements or adjustment of the measurement strategies ('model improvement' and 'adjusting measurement strategy' arrows).**

*Section 2.3: Line 195: Maybe add paper by Bloom and Williams (2015) and latest CLM paper by Post et al. (2017)?*

Added the Post et al. (2017) reference; the paper by Bloom and Williams (2015), although it is also a model-data fusion study, has a slightly different focus insofar as it uses ecological 'common sense' constraints and may not fit that well to the context here.

*Line 200: maybe worth adding ". . .same cost function value at the minimum"?*

Added as suggested.

*Line 203: Could add Thum et al. (2017) here.*

Added as suggested.

*Section 3.1: Worth mentioning that observation errors in a DA system should include the models errors, and what could give rise to errors in the model?*

We have included the following short paragraph on model errors:
**These off-diagonal elements are usually hard to specify, but they are important to quantify in a data assimilation system because they have considerable impact on the solution because of their influence on the weight of the respective observations in the cost function.**
**In addition to the observational errors, models also have errors, which, in a data assimilation system, are usually included in the observation errors. These errors in dynamical models are mainly caused by process parameterizations (instead of resolving the process), and by the discretization of analytical dynamics into a numerical model. A more detailed description of the different model error sources is given in Scholze et al. (2012).**

*Section 3.2: This section is very focused on Europe. It would be worthwhile detailing efforts that are underway in other regions, e.g. example the NASA Carbon Monitoring System (http://carbon.nasa.gov). This section also feels a little out of place. I might suggest incorporating it into the introduction to Section 3 or having it as a perspectives section at the end of the article.*

We have removed this section and moved a shortened version of the text to the Conclusions to give a perspective on operational monitoring systems at the end of the article as suggested, see also answer to the first comment above.

*Section 3.3: Lines 306-308: worth pointing out that using level 2 products may increase the observation uncertainty, particularly given parameters/processes implemented in retrieval algorithm may not be consistent with corresponding equivalent parameters/processes in the underlying model (and that this may be a benefit of using level 1 products – e.g. Quaife et al., 2008). Also perhaps worth explaining that for vegetation activity that VIs are an intermediate step in that they are "lower order" products – i.e. they are raw radiances but also do not require a complex retrieval algorithm; instead they require an atmospheric transport model and limited calculations.*

We are not quite sure what the referee is referring to here. The study by Quaife et al. (2008) is based on Level 2 data and not Level 1. Nevertheless, we have included a sentence on the possible inconsistencies when using Level 2 data.

**However, there is the risk that when using level 2 or higher products the parameters/processes implemented in the retrieval algorithm may not be consistent with corresponding equivalent parameters/processes in the underlying model, and thus cause additional errors in the assimilation.**

*Line 316-318: worth including that NDVI has also been used (e.g. MacBean et al., 2015a), and the advantages/disadvantages of using Vis.*

We have included the reference and referred to Section 3.3.2 (where we describe the disadvantages of such VIs over a physically based quantity such as FAPAR.) for a discussion on the difference between VIs and FAPAR.

**…and recently developed products based on biogeochemical processes, such as sun-induced fluorescence (SIF). There is also a range of remotely sensed vegetation indices, of which NDVI and leaf area index (LAI, e.g. Liu et al., 2014) (which is closely related to FAPAR) are examples. Both have been used in data assimilation studies: an example for NDVI is given by MacBean et al. (2015) and for LAI by Luke et al. (2011) and Barbu et al. (2014). In Section 3.3.2 we detail the difference between NDVI and FAPAR, and explain that FAPAR is based on physical principles. FAPAR has already been demonstrated to provide a strong constraint…**

*Line 318: Forkel et al. (2015) is another example of the use of FAPAR with a terrestrial model.*

Included the Forkel et al. (2014) reference.

*Line 321: and by optimizing parameters related to phenology and photosynthesis (MacBean et al., 2015b).*

We prefer not to cite a conference presentation.

*Line 322: Saying "Also assimilation of XCO2" comes a bit out of the blue here as you have just been talking about vegetation activity. Please could you say what is meant by XCO2, or refer to section 3.3.1.*

Changed the beginning of the sentence and added a reference to section 3.3.1:
**Remotely sensed atmospheric $CO_2$ concentration ($XCO_2$, see Section 3.3.1) has also been assimilated into a diagnostic terrestrial carbon cycle model to derive net $CO_2$ fluxes consistent with independent in situ measurements of atmospheric $CO_2$ and to reduce posterior uncertainties in the inferred net and gross $CO_2$ fluxes (Kaminski et al., 2016).**

*Lines 325-332: Other examples of the impact of soil moisture (and LAI and FAPAR) data assimilation on LAI and C fluxes include the work at CNRM with the ISBA-A-gs model, e.g. Barbu et al. (2014).*

Included the Barbu et al. (2014) reference:

**Remotely sensed atmospheric CO$_2$ concentration (XCO$_2$, see Section 3.3.1) has also been assimilated** into a diagnostic terrestrial carbon cycle model to derive net CO$_2$ fluxes consistent with independent in situ measurements of atmospheric CO$_2$ as well as to reduce posterior uncertainties in the inferred net and gross CO$_2$ fluxes (Kaminski et al., 2016). **Barbu et al. (2014) assimilated both soil moisture and LAI data into a land surface model, but their focus was on improving the hydrological and land surface physical quantities and not the carbon cycle.**

*Line 334: several studies have demonstrated the added benefit of aboveground biomass, including articles already cited (Richardson et al., 2010; Williams et al., 2005; Keenan et al., 2012). Might be worth listing a few examples, or, combining this section with aforementioned examples of C cycle related DA studies (section 2.3).*

We have added some references:
**So far, remotely sensed biomass data have not been used in carbon cycle data assimilation studies, but several studies (e.g. Richardson et al., 2010; Keenan et al., 2012; Thum et al., 2017) have demonstrated the added value of in situ above-ground biomass observations in constraining the terrestrial carbon cycle.**

*Line 340: LAI has been used in C cycle DA (see Barbu et al., 2014). Further to my comment on VIs above, perhaps it would be worth explaining somewhere in the text the differences between using VIs, FAPAR and LAI, why one would use one vs another?*

See answer above to comment Line 316-318.

*Line 341: Worth mentioning the dataset of Li et al. (2011) that has been used in several studies investigating trends in biomass. In fact, I expect that VOD data will be increasingly widely used for optimizing biomass in terrestrial biosphere models, and therefore I would suggest adding a discussion of what these data actually represent in Section 3.3.5 (i.e. how reliably can you estimate biomass (leaf or total aboveground?) from what is essentially a measure of water content).*

See answer below to comment Line 675.

*Line 343: I think that LST might be used in a similar manner to soil moisture in DA in the future, and not just as an input/boundary condition. Therefore perhaps it can be included with VOD in this context?*

Changed as suggested:
**However, these products are rather used as input or boundary conditions for terrestrial carbon cycle models (burned area and land cover) or, in the case of land surface temperature and vegetation optical depth, they have so far not been used in carbon cycle data assimilation studies.**

*Section 3.3.2 Lines 477: you mean significant difference in the absolute magnitude between the products (as the temporal and spatial patterns are quite consistent, as you state)? This was also a conclusion drawn by D'Odorico et al. (2014) and, to some extent, Tao et al.*

*(2015); therefore, it is worth mentioning that these studies agree on this point. As mentioned above, here or elsewhere I think it would be beneficial to have a discussion of the use of VIs and LAI. Arguably LAI is the variable that is most closely linked to standard terrestrial model state variables, therefore the reader should understand why one might use any of these three options for optimizing vegetation dynamics/activity, and the advantages/disadvantages of each. For example, if a modeler is mostly concerned with optimizing the overall magnitude of vegetation activity, careful choice of which FAPAR product to use is important (likely the same for LAI). If they are more concerned with temporal dynamics, one could argue that using a normalized lower order product (e.g. NDVI) that does not require such a sophisticated retrieval algorithm might be more appropriate. Perhaps you do not agree! But in any case, a discussion would be useful here. At the end of this section there is a particular focus on the JRC-TIP FAPAR product as opposed to one of the others, MODIS for example. It would be good to explain the reason for this choice, or to see more information on some of the other commonly available products.*

We have emphasised that the difference between the products lies mainly in the absolute magnitude and added that D'Odorico et al. (2014) and Tao et al. (2015) came to the same conclusion:

**Pickett-Heaps et al. (2014) concluded that although all six evaluated products display robust spatial and temporal patterns there is considerable disagreement in the absolute magnitude amongst the products and none of the products outperforms the others. This has also been confirmed by the studies of Dodorico et al. (2014) and Tao et al. (2015).**

We extended the discussion on the difference between Vis and FAPAR by adding LAI into it and mentioning the shortcomings of VIs and LAI compared to FAPAR:

**These indices generally exhibit some improvement in one respect but at the expense of degradation in another respect. Pinty et al. (2009) demonstrate the limitations of such VIs in representing the complex radiative properties of the canopy-soil system over the visible to NIR albedo range. Satellite-derived LAI products (e.g. Liu et al., 2014) seem to be an alternative to VIs. LAI is, however, model-dependent, i.e. the correct interpretation of this variable depends on the formulation of the model used in the retrieval scheme, and may differ from the interpretation adopted by the land biosphere model used for assimilating the LAI product (Disney et al., 2016).**
**A rational approach to addressing all these issues together is to design a physically-based quantity which is determined by the state of the canopy-soil system.**

The reason why there is a slight focus on the JRC-TIP product is mentioned in this section 3.3.2. We made this clearer now in the manuscript at two places:

**The JRC-TIP (Pinty et a., 2007) is an inverse modelling system that was explicitly designed to retrieve a set of land surface variables, including FAPAR, in a form that is compliant with the requirements for assimilation into terrestrial biosphere models, hence we focus in the following on this product.**

**TIP uses observed broadband albedo in the NIR and visible spectral domains as input. The prior information used in the retrieval is constant in space and time, i.e. all variability is determined from space (Kaminski et al., 2017). This is in contrast to other**

**retrieval approaches, which are based on prescribed land cover maps (e.g. Liu et al., 2014).**

*Lines 484-485: Please could you be clearer how the products in this sentence link to Table 2? JRC MGVI is not described in Table 2 for example.*

The JRC MGVI product is included as a footnote in Table 2 because it uses the same algorithm as used for the SeaWiFS product. We clarified this in the footnote and explicitly mention now JRC MGVI.

*Section 3.3.4 Lines 489-491: Although I appreciate you do not wish to provide an exhaustive description of retrieval algorithms, I think it would be helpful to qualitatively describe the difference between passive and active retrieval algorithms in one or two sentences here, as well as the fact different algorithms may produce either volumetric water content (absolute values) vs relative soil moisture values. I would be interested to see a discussion of GRACE land water content in this section.*

We included a short description on the retrieval of soil moisture from active instruments and added a short sentence on GRACE. We did not include a discussion of GRACE land water measurements here because they reflect the amount of ground water. This is different to the plant available soil moisture used in terrestrial ecosystem models and relevant for simulation of the terrestrial carbon cycle:
**Both passive radiometer systems, measuring the emitted microwave radiance ('brightness 590 temperatures'), and active radar systems, measuring backscattered microwave radiance, can be used to retrieve soil moisture. Various approaches exist that convert brightness temperatures and backscatter measurements into estimates of soil moisture, including radiative transfer model inversion approaches (e.g. Kerr et al. 2012, Owe et al. 2008), neural networks (e.g., Rodríguez Fernández et al. 2015), linear regressions (e.g., Al-Yaari et al. 2016), and change detection methods (Wagner et al., 1999). The latter is commonly applied to scatterometer measurements and yields, in contrast to the other approaches which provide soil moisture as volumetric water content, soil moisture as a percentage of total saturation.**

**Only Synthetic Aperture Radar is able to provide much higher spatial resolutions, up to a few tens of meters, yet at the cost of the revisit times. Also observations made by the Gravity Recovery and Climate Experiment (GRACE; Rodell et al. 2009) are sensitive to soil moisture, but the estimation of soil moisture content from these observations is not straightforward because they are also sensitive to changes in snow, surface water, groundwater, and vegetation.**

*Section 3.3.5 Line 670: do not need to reiterate what an active sensor is here.*

Removed the half sentence on what an active sensor is.

*Line 675: I see you do refer to the VOD product of Liu et al. here. Still, I think it would be beneficial to detail that this is based on VOD data and describe briefly how VOD are derived (following on from the mention of VOD in the soil moisture section) and how biomass is*

*estimated from VOD and their expected use/value for optimizing biomass (as discussed above), as well as for better understanding discrepancies in other sources of biomass data that you discuss towards the end of Section 3.3.5.*

We have added that the Liu et al (2015) AGB product is based on VOD. We did not include a discussion on how VOD is derived; that would be outside our scope.
**Furthermore, the emphasis is on the AGB of forests, although a global data set of AGB in all biomes for the period 1993-2012 has been produced based on** VOD data from **global passive microwave sensors, hence with spatial resolution of 10 km or coarser (Liu et al., 2015).** The AGB product is derived from a regression of VOD against observations of AGB from ground-based inventory data.

*Lines 676-684: updated reference: Santoro et al. (2015) – update to aforementioned papers providing biomass estimates across a wider range of biomes in the northern hemisphere.*

Updated the reference.
Santoro et al. (2015) provide a high resolution dataset (0.01°) over the northern hemisphere with a relative RMSE against National Forest Inventory between 12% and 45%.

*Line 711: Could you provide the biomass limit that the P-band BIOMASS mission will be able to resolve (to compare with the NISAR mission)?*

We added the following text to provide a biomass limit from P-band:
**The ESA BIOMASS mission (European Space Agency, 2012), to be launched in 2021, is a P-band radar that will provide near global measurements of forest biomass and height.** Measurements from airborne sensors indicate that even in dense tropical forests affected by topography, the P-band frequency used by BIOMASS will give sensitivity to biomass up to 350-450 t/ha (Minh et al., 2014; Villard and Le Toan, 2015).

*Given you mention the international soil moisture network in Section 3.3.4, it may be worth mentioning the international tree ring data bank in this section (https://data.noaa.gov/dataset/international-tree-ring-data-bank-itrdb), as these data represent a promising new direction for optimizing biomass across a range of biomes.*

Included as suggested, we added the following sentence at the end of Section 3.3.5:
As well as limitations caused by mission lifetimes, satellite measurements of biomass are unlikely to be sensitive enough to measure biomass increment except in rapidly growing plantations and tropical forests. Hence an important ancillary dataset for studies aiming to relate biomass to climate and environment is tree ring data (https://www.ncdc.noaa.gov/data-access/paleoclimatology-data/datasets/tree-ring).

*Minor comments and typos Line 252: maybe "between" better than "among"? Line 309-310: sentence could be simplified Line 111: benchmark Line 135: measurement Line 145: knowledge Line 234 and 241: related Line 255: diagonal Line 311: terrestrial Line 315: biogeochemical Line 420: that than Line 440: reflectance Line 576: complementarily Line 727: satellite Line 1395: Updated Thum et al. (2016) reference – see below.*

All corrected.

**Comments by referee 2, Thomas Kaminski:**

*1. Model-data fusion and data assimilation: L 37 states that both terms mean the same. Is this true? If yes, I suggest to keep one of the two for the rest of the manuscript instead of switching between the two. If not – maybe because by model-data fusion we could also understand some blending of observations with pre-computed model output – then be more precise in the definitions here and below use the appropriate term depending on context.*

We clarified the terminology by adding the following sentence and used 'data assimilation' throughout the manuscript.

**The term model-data fusion is sometimes understood in a more general way, where observational data is blended with (pre-computed) model output, whereas the term 'data assimilation' refers to a robust mathematical framework for improving model predictions with observational data.**

*2. L 47: "new observation": maybe rather "new data stream" or "new type of observation"*

Changed as suggested.

*3. L64: "In contrast to Ciais et al. (2014), who focus on carbon-cycle observations, we focus here on any kind of relevant observational data to be (potentially) assimilated in a terrestrial carbon cycle data assimilation system (CCDAS). In a CCDAS the observations are used to constrain the underlying model (i.e. to move model output quantities closer to the observations and reduce their posterior uncertainties) usually by parameter optimisation." This formulation could be improved. "any kind of" could be dropped in the first sentence, and the second sentence could read (for example): In a CCDAS non-carbon observations can be exploited to constrain the simulated carbon cycle indirectly through the relations implemented in the process model. Such observational constraints act by ruling out combinations of the unknowns in a CCDAS (typically a combination of process parameters, initial- or boundary conditions) that are inconsistent with the observations and thereby reduce uncertainties in the simulated output."*

Changed as suggested.

*4. L73: "Our focus lies on the terrestrial carbon cycle, because of the higher spatial and temporal variability in the net exchange fluxes and their associated higher uncertainties than form the ocean and anthropogenic components." Maybe not true on all relevant scales. Maybe just drop the sentence, no need to justify the terrestrial focus in this context.*

Changed as suggested.

*5. L101: in fact the weighting is in inverse proportion of the uncertainty, also appears below where Eq 1 is described*

Corrected.

*6. L103: either is maybe not appropriate?*

Indeed, removed 'either'.

*7. L118: Maybe you want to put: "Here, we follow the notation as introduced by Rayner et al. (2016)" at the beginning of the subsection, i.e. before you start using their notation.*

We included the reference to Rayner et al. (2016) in the first sentence of the subsection:
**The general problem of data assimilation can be formulated (following the notation of Rayner et al. [2016]) as follows:…**

*8. As we are dealing with assimilation of "multiple data streams" you could mention that usually each data steam requires its own observation operators, and in fact already here refer to Kaminski and Mathieu (2016/7), maybe even their octopus Figure. And for Eq 1 you could say that, for convenience of notation, now you combine all of them into a single H().*

Added two sentences to clarify this point. The first one in the second paragraph of this section:
**A data assimilation system consists of three main ingredients: a set of observations, a dynamical model including the observation operator, and an assimilation method. When assimilating multiple data streams each data stream usually requires its own observation operator (see e.g. Kaminski and Mathieu, 2016).**

And a second one after Equation (1):
**When multiple data streams with different observation operators are assimilated there will be several summands of the form of the second term on the right hand side of Equation (1), one for each data stream.**

*9. L122: Maybe drop this sentence. In fact the state are mixing ratios.*

Changed as suggested.

*10. L133: "thus evolves"?*

Changed the corresponding sentence to:
**We distinguish two basic approaches in data assimilation: sequential assimilation, which assimilates observations subsequently at discrete model time-steps, and variational assimilation,…**

*11. L136: "optimality" maybe you can find a better word? Maybe "adequacy"?*

Indeed, changed as suggested.

*12. Figure 1 is confusing in several respects (prior info enters cost, inner/outer loop to be confused with NWP terminology, U(o) not necessarily only a model output, cost function at*

*minimum does not imply availability of A, etc ...). Maybe you just want to drop it with the two sentences that describe it?*

We keep Figure 1 because referee #1 found it useful, but we updated the figure and changed the wording (inner/outer loop) in the text and caption to not be confused with NWP.

*13. L113-150 starting with "From Equation ..." could also be clearer, shortened or dropped (It does not follow from Eq. 1 that uncertainties are to be taken into account, but Eq. 1 follows from combining PDF descriptions of prior, observations, and model with a few simplifications, Mean and variance are not sufficient to characterise a multi-variate Gaussian, ...) Same holds for next paragraph ("assimilation problem is Gaussian" does not make sense; division by the matrix B is not straightforward...) Maybe just explain variables in Eq. 1 and then directly move to the paragraph starting with Rayner et al. (2016).*

Dropped the whole paragraph as suggested.

*14. L161: The "and" between citations is missing ("citep" would have worked for multiple citations), same problem occurs a few times further down below.*

Changed.

*15. L205: what about non-linear observation operators?*

Included also 'observation operators' in the text:
**...; in the case of non-linear models or non-linear observation operators only the simultaneous assimilation...**

*16. Section 3.1: It is good to introduce the different forms of errors. It would also be instructive to provide definitions of precision and accuracy.*

We think a definition is not needed here (and can be looked up in a dictionary if needed), it is more important that the terms are related to random and systematic errors, which we have done at the end of the respective bullet points.

*17. L242: Is is worth mentioning that the scale at which we trust the model may be larger than a grid cell?*

Changed as suggested:
**For instance a quantity simulated by a model is 'representative' for a given spatial and temporal resolution of the model grid. In fact, the scale at which we trust a model may be larger than a grid-cell.**

*L247: "In the case of satellite-based observations the representation error also includes errors in inferring a biophysical quantity from the photons measured at the sensor. We come back to this issue later." I would think that such errors in the retrieval rather go into the above two categories?*

Indeed this may be misleading and we have removed this sentence here.

*18. L 255: "they affect the prediction of the optimal solution in the same way as" could maybe be replaced by "they have considerable impact on the solution. This is because of their influence on the weight of the respective observations in the cost function." It is very good you stress this point. In fact you should take it up in the presentation of each data stream. So far it is only addressed in the XCO2 and the biomass sections.*

Changed the sentence as suggested. We did not mention this for each data stream explicitly again because here it is mentioned in general and not related to a particular data stream.

*19. L 263: What is inhomogeneous variance?*

Here, it means that the variance of each of the superposed Gaussian distributions is not the same.

*20. Section 3.2: Maybe add reference/web page of ICOS? Is is worth mentioning similar programmes outside Europe? "The measurements are designed": maybe better "the network" or "the observing system"?*

Added URL for ICOS web page and changed as suggested.

*L282: Paragraph may fit better into the beginning of section 3.3. Where you discuss the relevant observations provided by the sentinels, you are using our current perspecitve, i.e. S1-5. You could make this clear, because in a few years time are reader could wonder why you don't mention observations by S6 ... etc...*

This section has been removed and parts of it are now in the Conclusions sections, see also answer to first comment from referee #1.

*21. L 305 and 310: On L305 you write EO, then Earth Observation, then EO... something to be checked throughout ...*

After introducing EO in the beginning of the manuscript we now consistently use EO throughout the manuscript.

*22. L344: For example Luke (2011) assimilates LAI.*
We have changed this section (see answers to referee #1) and included the reference to Luke (2011).

*23. Section 3.3.1: Is it worth to briefly explain how a total column value can be sensitive or insensitive to surface fluxes?*

The following has been added:
**In the following we focus the discussion on sensors that have already delivered multi-year XCO$_2$ and XCH$_4$ data sets, i.e. SCIAMACHY and TANSO.**

**These satellite-derived XCO$_2$ and XCH$_4$ data products are sensitive to surface fluxes because CO$_2$ and CH$_4$ emission and uptake by surface sources and sinks results in the largest changes of the atmospheric CO$_2$ and CH$_4$ mixing ratio close to the Earth's surface and therefore modifies the observed vertical columns. This results in local or regional atmospheric enhancements (e.g., Buchwitz et al., 2017, discussing localized methane sources) or large-scale atmospheric gradients (e.g., Reuter et al., 2014, discussing CO$_2$ uptake by the terrestrial biosphere).**

*L400: ")" should be "("*

Corrected.

*L413: You could mention how the aggregation of errors to the 5 degree grid was performed.*

The following has been added:
**Each 5°×5° monthly grid cell also contains an estimate of the overall uncertainty (also shown in Fig. (3)) which has been computed taking into account random and systematic error components. The grid-cell uncertainty is computed from two terms: (i) using the reported uncertainties as given in the Level 2 (individual ground pixel) product files for each of the used satellite products (using an ensemble of SCIAMACHY and GOSAT Level 2 products) and (ii) using a term accounting for potential regional/temporal biases as obtained from validation using TCCON ground-based data (see above). The first term depends on the number of individual observations added (the error reduces in proportion to the square root of the number of observations added) whereas the latter term is constant and in the range 0.57 – 0.87 ppm depending on satellite XCO$_2$ product or in the range 6 – 10 ppb for XCH$_4$.**

*L435: Maybe update reference to latest version of the CCI CAR.*

Done.

*L439: Is is worth mentioning planned XCO2 missions?*

Note that additional missions, not discussed in detail in our manuscript, are already mentioned. Therefore, we have added here planned missions only shortly by adding the following:
**…China's TanSat (planned launch at the end of 2016), which will deliver XCO2 with similar characteristics to NASA's OCO-2. It can be expected that future satellites will provide improved measurements, in particular with respect to more localized emission sources (e.g., Bovensmann et al., 2010; Buchwitz et al., 2013; Ciais et al., 2015).**

*24. Section 3.3.2: L445: remove one ")".*

Corrected.

*L458: "closely follows the state of the vegetation" could be "is determined by the state of the canopy-soil system"*

Changed as suggested.

*L475: Disney et al. (2016) also compare two products.*

Included the Disney et al (2016) reference here:

**McCallum et al. (2010) looked at four FAPAR data sets over Northern Eurasia for the year 2000, Pickett-Heaps et al. (2014) evaluated six products across Australia, D'Odorico et al. (2014) compared three products over Europe, Tao et al. (2015) assessed five products over different land cover types, and Disney et al. (2016) compared two FAPAR products derived from GlobAlbedo and MODIS data.**

*L 480: To simplify the sentence maybe move the part in brackets up to the definition of L460.*

Done as suggested, the text with the definition of FAPAR now reads:

**The Fraction of Absorbed Photosynthetically Active Radiation (FAPAR), which is a normalised fraction with values ranging from 0 to 1, provides information on the photosynthetic activity of the land vegetation.**

*L503: "see 2" should be "see table 2".*

Corrected.

*Regarding correlation of uncertainty you might add on L506: after "periods.": To reduce disk space, by default, JRC-TIP products are delivered without correlations among the uncertainties between individual variables, even though these correlations are available. An estimate of uncertainty correlation in space and time is not provided. The JRC-TIP products derived from MODIS (collection 5) broadband albedos minimise temporal uncertainty correlation as each collection 5 albedo value is derived as integral exclusively of observations over non-overlapping 16-day periods.*

Changed as suggested, except for the last sentence (which seems to be too specific).

*25. Section 3.3.3: L 514: "directly related" In the context of data assimilation, is it worth mentioning that there are complex processes which require complex models as observation operators for SIF, in order to extract the maximal benefit from this data steam?*

Included a sentence on this at the end of the paragraph:

**But in the context of DA and in order to extract the maximal benefit from SIF data, the complex processes responsible for SIF in the plants' photochemical systems (as mentioned above) require complex models as observation operators for SIF.**

*L 524: "lies" could be "relies"?*

Changed as suggested.

*L 530: Why is the simplicity of the forward model related to the fact that least squares is applied, which might also work with complex models?*

Changed to:
**The retrieval forward model is thus simple and can be linearised (e.g. Guanter et al., 2012; Köhler et al., 2015}, which simplifies the inversion**.

*L 540: Does "a compromise" make sense here? Isn't it rather the definition of the grid size that is determined by a compromise and the number of retrievals then just a function of this choice of grid size (plus the other factors mentioned)? When discussing the spatial and temporal sampling, it might be instructive to mention the variability in time as you do it in space.*

Typically, the smallest spatial or temporal grid size is defined as a function of the application and region of interest (if not global studies are considered), which then defines the other dimension (temporal and spatial), and the number of retrievals to be considered as the reviewer is pointed out. A short clarification has been added to the text.
**The number of retrievals to be aggregated into a given grid-cell results from a compromise between spatial resolution, temporal resolution and precision of the gridded product, the size of the spatial and temporal bins being exchangeable in terms of their effect on the random uncertainty.**

*26. Section 3.3.4: L600: "cost of the": maybe better "cost of long"*

Changed as suggested.

*L652: ",5."*

Corrected, should read 'see Figure 5'.

*L658: "to improve the model's hydrology" in fact in a CCDAS we are after the indirect constraint on carbon, so this restriction may not be needed here?*

*Indeed, we have slightly changed the text:*
**…that allows for a systematic assimilation into land surface models. These products have been used to improve model hydrology by, for example, Martens et al. (2016) who showed that…**

*27. Conclusions: L724: "observational characteristics of the observational data" maybe you meant error or uncertainty characteristics?*

Yes, corrected.

*L730: "correlations": "uncertainty correlations" or "error correlations"*

Corrected to 'error correlations'.

*L732: "For example, while FAPAR data constrain mainly the phenology component of a terrestrial carbon cycle model, soil moisture data, in contrast, constrain the hydrological component," see above regarding the indirect constraints. You probably do not need to write this. In fact FAPAR can provide an important constraint on hydrology (see Kaminski et al., 2012)*

Removed this sentence as suggested, this also increases the readability of this section.

*28. Fig 3: I'd suggest to go for a 6 panel figure, the four maps are tiny; Better use degree symbol in capton.*

We have improved the figure along the lines suggested and for the revised version of the manuscript we use the degree symbol in the caption.

*29. Table 1: I suggest to replace "parameter" by "variable"*

Changed as suggested.

*30. Table 4: You could add wave lengths to the bands, for many colleagues the band names don't mean anything.*

We do not think that adding the wave lengths provides important additional information in the context of the paper here, so we kept the table as is.

*There are quite a number of typos. Many of them (e.g. "Reflectamce-based" or "assessemt" or "observeing") can be detected by a spell checker.*

All corrected.